# Temporal Transcriptome Dynamics of *Longissimus dorsi* Reveals the Mechanism of the Differences in Muscle Development and IMF Deposition between Fuqing Goats and Nubian Goats

**DOI:** 10.3390/ani14121770

**Published:** 2024-06-12

**Authors:** Yuan Liu, Xianfeng Wu, Qian Xu, Xianyong Lan, Wenyang Li

**Affiliations:** 1Fujian Provincial Key Laboratory of Animal Genetics and Breeding, Institute of Animal Husbandry and Veterinary Medicine, Fujian Academy of Agricultural Sciences, Fuzhou 350013, China; liuyuan@faas.cn (Y.L.); wuxianfeng@faas.cn (X.W.); xuqian@faas.cn (Q.X.); 2College of Animal Science and Technology, Northwest A&F University, Xianyang 712100, China

**Keywords:** goat, transcriptome, growth, *Longissimus dorsi* muscle, development, genetics

## Abstract

**Simple Summary:**

The molecular mechanism underlying muscle development in goats is complex, and the key genes commanding the growth performance and meat quality of different breeds remain unclear. Herein, we present temporal transcriptome research on *Longissimus dorsi* (LD) muscle of two goat breeds from embryo to 12 months old, with extreme phenotypic differences in production and meat quality traits, using an Illumina system for the identification of key genes to improve production efficiency in goats.

**Abstract:**

In this study, we measured the growth performance and intramuscular fat (IMF) content of the *Longissimus dorsi* (LD) of Fuqing goats (FQs) and Nubian goats (NBYs), which exhibit extreme phenotypic differences in terms of their production and meat quality traits. RNA-Seq analysis was performed, and transcriptome data were obtained from the LD tissue of 3-month fetuses (E3), 0-month lambs (0M), 3-month lambs (3M), and 12-month lambs (12M) to reveal the differences in the molecular mechanisms regulating the muscle development and IMF deposition between FQs and NBYs. The results showed that a higher body weight and average daily gain were observed in the NBYs at three developmental stages after birth, whereas a higher IMF content was registered in the FQs at 12M. Additionally, transcriptome profiles during the embryonic period and after birth were completely different for both FQs and NBYs. Moreover, DEGs (*KIF23*, *CCDC69*, *CCNA2*, *MKI67*, *KIF11*, *RACGAP1*, *NUSAP1*, *SKP2*, *ZBTB18*, *NES*, *LOC102180034*, *CAPN6*, *TUBA1A*, *LOC102178700*, and *PEG10*) significantly enriched in the cell cycle (ko04110) at E3 (FQs vs. NBYs), and DEGs (*MRPS7*, *RPS8*, *RPL6*, *RPL4*, *RPS11*, *RPS10*, *RPL5*, *RPS6*, *RPL8*, *RPS13*, *RPS24*, *RPS15*, *RPL23*) significantly enriched in ribosomes (ko03010) at 0M (FQs vs. NBYs) related to myogenic differentiation and fusion were identified. Meanwhile, the differences in glucose and lipid metabolism began at the E3 timepoint and continued to strengthen as growth proceeded in FQs vs. NBYs. DEGs (*CD36*, *ADIROQR2*, *ACACA*, *ACACB*, *CPT1A*, *IGF1R*, *IRS2*, *LDH-A*, *PKM*, *HK2*, *PFKP*, *PCK1*, *GPI*, *FASN*, *FADS1*, *ELOVL6*, *HADHB*, *ACOK1*, *ACAA2*, and *ACSL4*) at 3M (FQs vs. NBYs) and 12M (FQs vs. NBYs) significantly enriched in the AMPK signaling pathway (ko04152), insulin resistance (ko04931), the insulin signaling pathway (ko04910), fatty acid metabolism (ko01212), and glycolysis/gluconeogenesis (ko00010) related to IMF deposition were identified. Further, the results from this study provide the basis for future studies on the mechanisms regulating muscle development and IMF deposition in different breeds of goats, and the candidate genes identified could be used in the selection process.

## 1. Introduction

China has abundant and varied genetic resources for goat (*Capra hircus*) breeding, and the Fuqing goat (FQ) is one of the local excellent breeds raised in Fujian Province, China [1]. The meat of FQs is more popular with local consumers because of its appropriate intramuscular fat content (IMF, more than 3.0%) [2], which is a major meat quality trait that influences aroma, tenderness, and juiciness [3]. However, the listed weight of FQs is less than 25 kg at 12 months old, and they do not produce much meat [4]. In contrast, Nubian goats (NBYs), which were introduced to China from the Nubian region of Africa, grow much faster and are used for the genetic improvement of local goat breeds in southern China [5]. Our previous work confirmed that the phenotypes of growth and meat quality were significantly different between the two breeds and that NBYs have a greater body weight and lower IMF content than FQs [2].The extreme differences in growth and meat quality traits between FQs and NBYs provide us with an ideal model for studying goat production, but the economic potential of their biodiversity remains underutilized due to a lack of knowledge of their genetic characteristics.

Growth performance is an economically important trait in goat production. Mass accounts for 40–60% of the body weight of mammals, and skeletal muscle is considered the most important tissue for producing meat livestock. Many studies have focused on the growth and development of skeletal muscle to identify relevant biological mechanisms underlying meat yield and quality and offer the possibility of improving growth traits in goats [6,7,8,9]. In addition, skeletal muscle development in mammals can be divided into prenatal and postnatal stages; the number of muscle fibers increases before birth and the volume of muscle fibers increases after birth [10]. In the embryonic stage, the main period of muscle formation involves skeletal muscle beginning with the mesoderm, from which the myoblasts first migrate to the limbs and trunk and proliferate rapidly [11]. When myoblasts, namely primary muscle fibers, reach a given number, they differentiate into slow or fast types to form complete skeletal muscle, also known as secondary muscle fibers [12]. The growth of skeletal muscle after birth in mammals depends mainly on the circumference and length of the muscle fibers [13]. In goats, the primary muscle fibers disappear and are completely converted to secondary muscle fibers between 90 days and 120 days of gestation [14]. In addition, the characteristics of skeletal muscle at the weaning stage, which is characterized by changes in feed intake, and at the age at market, which determines meat yield and quality, have also been studied by researchers of goats [15]. Since muscles form meat, it would be beneficial to know the molecular drivers that regulate their development and metabolism. Dynamics in gene expression profiles during the key developmental stages of muscle in goats are an essential consideration for future developments in commercial goat production.

RNA sequencing is a high-throughput sequencing method for characterizing gene expression in specific tissues that provides accurate counts of transcripts to measure relative expression and to discover new exons or genes [16]. This approach is widely used for research on gene expression profiles revealing muscle growth and development, and the transcriptome profiles of different breeds and muscle tissues are becoming increasingly clear in goats [6,7,8,17,18]. Some functional differentially expressed genes (DEGs) from different kinds of materials in KEGG pathways, such as the Jak-STAT signaling pathway [18], MAPK signaling pathway [19], and PI3K-Akt signaling pathway [7,20], have been found to play crucial roles in muscle growth and development in goats. However, comparative transcriptome analyses to identify genetic differences have focused on only a few goat breeds, and the key genes that control muscle development might differ among breeds. Most of the skeletal muscle transcriptome properties of goat breeds, especially indigenous goats growing in a specific area, are still unknown. The present study is therefore an attempt to obtain an overview of the skeletal muscle transcriptome of FQs and NBYs. The aim of this study was to compare the differences in gene expression between FQs and NBYs during the development of *Longissimus dorsi* (LD) muscles. Our findings provide insight into the relationship between dynamic changes in the skeletal muscle transcriptome and growth traits and reveal potential candidate genes with important roles in the regulation of muscle growth for molecular breeding in goats.

## 2. Materials and Methods

### 2.1. Ethics Statement

The experimental animals and procedures performed in this study were approved, with reference number 202207FJ002, by the International Animal Care and Use Committee of Fujian Academy of Agricultural Sciences (FAAS), Fujian Province, China. The care and use of experimental animals fully complied with local animal welfare laws, guidelines, and policies.

### 2.2. Animal Management and Sample Collection

The experimental animals used in this study were FQs and NBYs, which were maintained with a unified management system at the Fujian Academy of Agricultural Sciences (FAAS) goat experimental farm in Fuqing city. Purebred FQ (2–3 years old, 30–35 kg) and NBY (2–3 years old, 55–65 kg) ewes (*n* = 20 per breed) were uniformly estrused and mated to purebred rams of their respective breed. The 20 FQs and 18 NBYs of successfully mated ewes were artificially fed in accordance with the Chinese feeding standards for meat-producing sheep and goats (NY/T816-2004) [21] in different pens according to breed. During this period, the pens were kept dry and clean, and the ambient temperature was controlled at 20–30 °C. Forty-five (24 male, 21 female) and thirty-seven (17 male, 20 female) fetuses or lambs were obtained from FQ ewes and NBY ewes, respectively. At 75 days of age, all the lambs were separated from their mothers after weaning and housed in pens by sex and bred in accordance with the Chinese feeding standard for meat-producing sheep and goats (NY/T816-2004) [21].

The LD muscles of fetuses or lambs were obtained at four stages, including fetuses at 3 months (E3, F-E3, and N-E3), lambs at 0 months (0M, F-0, and N-0), 3 months (3M, F-3, and N-3) and 12 months (12M, F-12, N-12), respectively. At each sampling stage, we randomly selected three male fetuses or lambs from different ewes of each breed. There were three biological replicates in each group, and only one offspring was selected for each ewe. Finally, twenty-four muscle samples, which were divided into eight groups, were immediately snap-frozen in liquid nitrogen and stored at −80 °C until further use. 

### 2.3. Body Weight, Average Daily Gain, and IMF Collection

Body weight (BW) was recorded for all lambs at 0M, 3M, and 12M. Then, the average daily gain (ADG) of all the goats from 0M to 3M, 3M to 12M, and 0M to 12M was calculated. The IMF contents of all slaughtered individuals were determined using the Soxtec extraction protocol [22].

### 2.4. Total RNA Extraction, Quantification, and Qualification

Total RNA was extracted from 24 frozen LD tissues using TRIzol reagent (Life Technologies, Carlsbad, CA, USA) according to the manufacturer’s instructions. The RNA concentration and purity were measured using a NanoDrop 2000 (Thermo Fisher Scientific, Wilmington, DE, USA). RNA integrity was assessed using the RNA Nano 6000 Assay Kit of the Agilent Bioanalyzer 2100 system (Agilent Technologies, Santa Clara, CA, USA). Only high-quality RNA with an RNA integrity number (RIN) ≥ 7.0 could be subjected to subsequent procedures. 

### 2.5. RNA-Seq Analyses

A total of 1 μg of RNA per sample was used as input material for the RNA-Seq sample preparations. Sequencing libraries were generated using the Hieff NGS Ultima Dual-mode mRNA Library Prep Kit for Illumina (Yeasen Biotechnology (Shanghai, China) Co., Ltd.) following the manufacturer’s recommendations, and index codes were added to attribute sequences to each sample. Briefly, mRNA was purified from total RNA using poly-T oligo-attached magnetic beads. First-strand cDNA was synthesized, and second-strand cDNA was subsequently synthesized. The remaining overhangs were converted into blunt ends via exonuclease/polymerase activities. After adenylation of the 3′ ends of the DNA fragments, NEBNext adaptors with hairpin loop structures were ligated to prepare for hybridization. The library fragments were purified with an AMPure XP system (Beckman Coulter, Beverly, MA, USA). Then, 3 μL of USER Enzyme (NEB, Ipswich, MA, USA) was incubated with size-selected, adaptor-ligated cDNA at 37 °C for 15 min, followed by 5 min at 95 °C before PCR. Then, PCR was performed with Phusion High-Fidelity DNA polymerase, universal PCR primers, and Index (X) Primer. Finally, the PCR products were purified (AMPure XP system), and library quality was assessed on an Agilent Bioanalyzer 2100 system. Then, RNA libraries were constructed, and sequencing was performed by Beijing Biomarker Technologies (BMK) Co., Ltd. (Beijing, China) on an Illumina NovaSeq 6000 platform. All the original transcriptome sequencing data of 24 samples were uploaded to the NCBI database. The database entry number obtained is PRJNA1077611.

### 2.6. Bioinformatic Analyses

Raw data (raw reads) in fastq format were first processed through in-house Perl scripts. In this step, clean data (clean reads) were obtained by removing reads containing adapters, reads containing poly-N, and low-quality reads from the raw data. Moreover, the Q30, GC content, and sequence duplication level of the clean data were calculated. All downstream analyses were based on high-quality clean data.

The clean reads were then mapped to the reference genome sequence (Capra_hircus.ARS1.genome.fa), which was predefined for the analysis, using HISAT2 software (version 2.2.1) [23]. To optimize the annotation information of a genome, the discovery of novel transcripts and genes was achieved by StringTie (version 2.2.0) [24] on the basis of the reference genome. The mapped reads were assembled and compared with the original annotations of the genome. The transcript regions without annotations obtained by the above processes were defined as novel transcripts, excluding short transcripts (coding peptides with less than 50 amino acids) or those containing only one exon.

Novel genes (transcripts) were annotated by DIAMIND (version 4.x.) [25] against databases including the NR, Swiss-Prot, COG, KOG, and KEGG databases. KEGG Orthology data of the novel genes were obtained via the above processes [26]. GO analyses of the novel genes were performed using the underlying software InterProScan (https://www.ebi.ac.uk/jdispatcher/pfa/iprscan5, accessed on 17 January 2024) based on the InterPro database [27]. The amino acid sequences of the novel genes were subjected to BLAST searches against the Pfam database [28] by HMMER (version 3.4) [29] to obtain the annotation information. The StringTie Reference Annotation Based Transcript (RABT) assembly method was used to construct and identify both known and novel transcripts from the HISAT2 [23] alignment results.

### 2.7. Differential Expression Analyses

Gene expression levels were quantified as fragments per kilobase of transcript per million fragments mapped (FPKM) [30]. Differential expression analysis of the two groups was performed using DESeq2 (version 1.0). The resulting *p*-values were adjusted using Benjamini and Hochberg’s approach for controlling the false discovery rate. Genes with an adjusted *p*-value < 0.05 and a fold change ≥ 2 according to DESeq2 were considered to be differentially expressed genes (DEGs).

Pearson’s correlation analysis was applied in this project to evaluate the reproducibility of the biological replicates, and samples with a Pearson correlation coefficient (PCC) less than 0.8 were removed from the same group of samples for further analysis. Principal component analysis (PCA) was performed on the FPKM of each sample using g-models in R (version 3.1.3, https://cran.r-project.org/, accessed on 19 January 2024). The similarities among the samples were displayed by reducing the dimensionality of the data into three principal components.

Gene ontology (GO) enrichment analysis of the DEGs was implemented using the clusterProfiler (version 4.12.0) package-based Wallenius noncentral hypergeometric distribution [31], which can adjust for gene length bias in DEGs. We used the KOBAS [32] database and clusterProfiler to test the statistical enrichment of DEGs in KEGG pathways. The sequences of the DEGs were subjected to BLAST (BLASTx) against the genome of a related species (the protein–protein interactions of which were identified in the STRING database (http://string-db.org, accessed on 21 January 2024)) to obtain the predicted PPI of these DEGs. Then, the PPI of these DEGs were visualized with Cytoscape (version 3.10.0) [33].

### 2.8. Weighted Gene Coexpression Network Analysis (WGCNA)

The RNA-Seq data were analyzed to construct gene coexpression networks using the R package (version 4.0.0) weighted gene coexpression network analysis (WGCNA) [34]. On the basis of the software default standard, the coconstructed genes were screened from the FQs RNA-Seq data for analysis. The module eigengenes were intended to describe the most common gene expression models in each module. The module eigengenes summarized the module overview and feature data because they were the first major component of the expression matrix. Pearson’s correlation coefficients were used to calculate the correlation between the modular eigengenes and each treatment period of FQs. The correlation is reflected by the depth of the color in the heatmap.

### 2.9. Trend Analysis of DEGs 

To uncover gene expression trends of DEGs among the four timepoint groups between FQs and NBYs, a gene coexpression analysis was performed using k-means clustering and BMKCloud (https://www.biocloud.net, accessed on 22 January 2024) [35]. Then, KEGG and GO analyses of different *K* clusters were performed.

### 2.10. Validation by qPCR

To verify that the RNA-Seq results were legitimate, twelve genes were screened by qPCR. The total RNA used for qPCR was generated from the same RNA used for RNA-Seq. The primers were designed using Primer-BLAST (http://www.ncbi.nlm.nih.gov/tool/primer-blast/, accessed on 22 January 2024), and the primer sequences are shown in Appendix A. The primers were synthesized by Fuzhou Shangya Biotechnology Co., Ltd. (Fuzhou, China). qPCR was performed on a Light Cycler 480 II Real-Time PCR System (Roche, Basel, Switzerland). The reaction volume of 20 μL included 10 μL of 2X SYBR Green MasterMix reagent (Thermo Scientific, Waltham, MA, USA), 1 μL of cDNA, and 0.2 μL of each primer (10 μM).

The stable endogenous genes *GAPDH* [36], *ACTB* [37], *HSP90* [38], *PPIB* [39], *EIF3K* [39], *API5* [40], *DRAP1* [41], and *WSB2* [41] were selected for data normalization according to previous reports. The stability of the endogenous genes was analyzed using their RNA-Seq data (Appendix A). The data from three replicates of each sample were collected, the mRNA expression of the samples was normalized to that of *API5*, which was selected as the most stable endogenous gene, and the fold change was calculated using the 2^−ΔΔCT^ method.

### 2.11. Data Analysis

The statistical software used was SPSS (version 22.0) (IBM Corp., Armonk, NY, USA). Each replicate served as the experimental unit for all the statistical analyses. Independent sample *t*-tests were used to examine the statistical significance of the differences in gene expression, BW, ADG, and IMF between the two groups. The data from two groups were analyzed by Student’s *t*-test. Error bars show the standard error of the mean, ** *p* < 0.01, * *p* < 0.05. Pearson’s correlation analysis was performed to examine the relationship between the RNA-Seq and qPCR results, and the significance of the differences was examined by a t-distribution test. *p* < 0.05 was considered to indicate statistical significance. 

## 3. Results

### 3.1. Production Traits

The production traits of FQs and NBYs are shown in Figure 1. The main phenotypic traits related to meat production strongly differed between the two breeds, as expected. A higher BW and ADG were detected in the NBYs at three developmental stages after birth, whereas a higher IMF content was detected in the FQ goats at 12M. This means that IMF deposition in the FQ group, which started before 12M, occurred earlier than in the NBYs group, as the IMF content did not significantly (*p* > 0.05) differ between the NBY and FQ groups at 0M or 3M.

### 3.2. Summary of RNA-Seq Results

Appendix A summarizes the RNA-Seq results. A total of 24 samples were processed for transcriptome sequencing, generating 428.23 Gb of clean data. At least 15.85 Gb of clean data were generated for each sample, with a minimum of 94.54% of the clean data achieving a quality score of Q30 (94.54–95.65%). Clean reads from each sample were mapped to a specified reference genome. The mapping ratio of each sample against the reference genome ranged from 96.75% to 97.49%. In this study, we detected the expression of 34,186 mRNA transcripts in 24 LD samples, among which 20,593 genes were annotated for function and 13,593 were newly discovered genes. 

### 3.3. Cluster Analysis of 24 LD Muscle Libraries

To evaluate the reliability of the samples, PCC values were utilized as assessment indices for the correlation of biological repetitions in the 24 libraries (Figure 2A, Appendix A). Three samples (F-0-2, N-E3-1, and N-3-2), whose PCC values in the same group were 0.23–0.76, were clustered far from their replicated samples. These samples were excluded from our subsequent analysis. Overall, 21 samples could be divided into two major clusters, which was consistent with the muscle development stage. The first cluster contained breeds sampled during the embryonic period for both FQs and NBYs. The second cluster contained breeds sampled after birth. The PCA results were consistent with the PCC analysis results (Figure 1B). The first, second, and third principal components (PC1, PC2, and PC3) explained 37.1, 8.9, and 7.2% of the variance, respectively. The samples collected during the embryonic period and after birth could be divided by PC1. This result suggested that the difference in growth performance between FQs and NBYs after birth was confirmed by the large-scale transcriptome analysis.

### 3.4. Identification of DEGs at Different Growth Stages in FQs and NBYs

We compiled DEG profiles for the two different goat breeds at four stages with a *p*-value < 0.05 and a fold change ≥ 2 for each comparison group. In total, 2121 (973 up and 1148 down), 1263 (674 up and 589 down), 1170 (443 up and 727 down), 1787 (626 up and 1161 down), 6926 (3299 up and 3627 down), 1633 (985 up and 648 down), 103 (55 up and 48 down), 5373 (2615 up and 2758 down), 1617 (814 up and 803 down), and 83 (30 up and 53 down) genes were differentially expressed in F-E3 vs. N-E3 (Appendix A), F-0 vs. N-0 (Appendix A), F-3 vs. N-3 (Appendix A), F-12 vs. N-12 (Appendix A), F-E3 vs. F-0 (Appendix A), F-0 vs. F-3 (Appendix A), F-3 vs. F-12 (Appendix A), N-E3 vs. N-0 (Appendix A), N-0 vs. N-3 (Appendix A), and N-3 vs. N-12 (Appendix A). The most DEGs were obtained for F-E3 vs. F-0 and N-E3 vs. N-0, while the fewest DEGs were obtained for F-3 vs. F-12 and N-3 vs. N-12 (Figure 3). Interestingly, we found that fewer DEGs were obtained during the fetal and birth periods, and more DEGs were obtained after weaning between the two breeds. However, in the same breed, more DEGs were obtained between the fetal and birth periods, while the number of DEGs sharply decreased after weaning. This suggests that subtle transcriptomic differences between the two breeds during the embryonic period may play a decisive role in the phenotypic differences in postnatal growth traits.

### 3.5. GO and KEGG Enrichment Analyses of DEGs between FQs and NBYs 

#### 3.5.1. GO Enrichment Analyses

To better understand the differences in biological mechanisms among the four developmental stages, GO enrichment analyses of the DEGs in FQs and NBYs, which included biological process (BP), cellular component (CC), and molecular function (MF) enrichment, and the removal of redundant GO terms by REViGO (revigo.irb.hr, accessed on 23 January 2024), were conducted. The 1421, 1156, 1121, and 1365 DEGs at E3, 0M, 3M, and 12M between FQs and NBYs, respectively, were significantly enriched (*p*-adj < 0.05) in 429 (Appendix A), 523 (Appendix A), 516 (Appendix A), and 443 (Appendix A) different GO terms, which subsequently decreased to 145, 56, and 83, respectively, by removing the redundant GO terms using REViGO (Table 1).

At E3, we focused on the representative major GO terms (Table 2) and found that most of the GO terms were related to the cell cycle and cell division (GO:0051988, GO:0000281, GO:0034080, GO:0000786, GO:0000776, GO:0046982, and GO:0010997), which accounted for the differences in skeletal muscle fusion and development (GO:0014905, GO:0031674, and GO:0062023) between FQs and NBYs, and these differences were likely regulated by insulin-like growth factor (GO:0042567). In addition, some GO terms related to immunity and disease (GO:0019731) were significantly enriched.

Lambs face the challenges of environmental changes and changes in nutrient sources after birth, which stimulate the immune and antioxidant functions of the body, and the GO terms reflecting the differences between the two breeds were significantly enriched at 0M (Table 2). The significantly enriched GO terms were mainly related to immunity and diseases (GO:0006956, GO:0043277, and GO:0016667), and mitochondria-related pathways related to cellular energy supply organelles were also significantly enriched (GO:0034551 and GO:0006390). Skeletal muscle differentiation was largely complete, whereas development related to the functional role of the skeletal muscle system was still ongoing (GO:0030316 and GO:0046982).

After birth, the effects of the metabolic differences in the *Longissimus dorsi* muscle between FQs and NBYs on growth and meat quality traits were gradually revealed, mainly reflected in the differences in cholesterol metabolism (GO:0016126, GO:0045121, and GO:1990666), glucose metabolism (GO:0008281 and GO:0070991), and the muscular system (GO:0030017, GO:0045121, GO:0001968, and GO:0045112) at 3M (Table 2). Similarly, significantly enriched GO terms related to immunity and disease were still identified (GO:0032682, GO:0001783, and GO:0042025).

At 12M, the goats were mature, and the weight and tissue function of the skeletal muscle were basically stable. Most of the GO terms enriched by the DEGs in FQs and NBYs were related to skeletal muscle composition and function (GO:0045662, GO:0031430, GO:0005581, and GO:0003785), which also determined the differences in growth and meat quality traits between the two breeds (Table 2).

#### 3.5.2. KEGG Enrichment Analyses

To explore the biological functions of the DEGs, KEGG enrichment analysis was carried out. The 55, 76, 45, and 75 KEGG pathways were significantly enriched (*p* < 0.05) at the E3, 0M, 3M, and 12M timepoints (Appendix A), respectively. The top 20 significantly enriched KEGG pathways from each timepoint were selected for further analysis (Figure 4).

The Hippo signaling pathway (ko04390), which was enriched at E3 (Figure 4A), mainly controls organ size by regulating cell proliferation and apoptosis [42]. The enrichment in genes related to the cell cycle (ko04110) also further verified the differences in skeletal muscle cell division and proliferation between FQs and NBYs at E3. Moreover, rapid division and proliferation were the main characteristics of cancer cells compared with normal cells, and many cancer-related KEGG pathways (ko05206, ko05200, ko05226, ko05203, and ko05225) were significantly enriched. Moreover, for the GO enrichment results at the E3 timepoint, KEGG pathways (ko05322, ko04916, and ko04610) related to immunity and disease were screened. Finally, the enrichment in the regulation of lipolysis in adipocytes (ko04923) and steroid biosynthesis (ko00100) suggested that the difference in fat metabolism in the *Longissimus dorsi* muscle between FQs and NBYs started during the embryonic period.

The enrichment of a large number of immune- and disease-related KEGG pathways (ko05322, ko05150, ko04610, ko05340, ko04662, and ko04612) between FQs and NBYs was a major feature of the 0M timepoint (Figure 4B). Interestingly, nine of the top twenty enriched KEGG pathways (ko01212, ko04152, ko00640, ko04068, ko05230, ko04930, ko04922, ko04920, and ko04931) at 3M and 12M were consistent (Figure 4C,D), and these KEGG pathways were related to fat and glucose metabolism, which may explain the difference in the IMF content trait between FQs and NBYs.

#### 3.5.3. Protein Network Analyses

A protein–protein interaction (PPI) network analysis was performed for up- and downregulated DEGs to identify hub genes and biological pathways potentially affecting the differentiated phenotypes of FQs and NBYs. First, a PPI network was constructed from the STRING database. Moreover, the PPI network of the hub genes was constructed with cytoHubba from Cytoscape. In total, 1334 (629 up and 705 down), 952 (542 up and 410 down), 789 (270 up and 519 down), and 1299 (319 up and 980 down) genes identified at E3 (Appendix A), 0M (Appendix A), 3M (Appendix A), and 12M (Appendix A) were mapped to the PPI network between FQs and NBYs (Figure 5).

The 63 hub genes, which included four clusters, were screened in eight DEG sets (Figure 5I). The first hub gene cluster contained 22 genes (*ACAA2*, *ACACB*, *ACADL*, *ACADVL*, *ACOX1*, *ACSL1*, *ACSL4*, *ACSS3*, *CPT1A*, *CPT1B*, *EHHADH*, *FASN*, *HADHB*, *HMGCS1*, *HSDL2*, *MLYCD*, *PDK4*, *PLIN1*, *PNPLA2*, *PPARG*, *SQLE*, and *SREBF2*) involved in lipid storage and fatty acid oxidation processes. The second hub gene cluster contained 17 genes (*BPGM*, *HK2*, *MRPL11*, *MRPS7*, *MTIF2*, *PC*, *PFKP*, *PKM*, *RPL23*, *RPL4*, *RPL5*, *RPL8*, *RPS11*, *RPS13*, *RPS15*, *TIGAR*, and *UBC*) involved in ribosome functions and glucose catabolic processes. The third hub gene cluster contained 13 genes (*BIRC5*, *BUB1*, *BUB1B*, *CCNA2*, *CDC20*, *CDK1*, *CENPF*, *KIF11*, *MELK*, *NCAPG*, *NUSAP1*, *PBK*, and *TTK*) involved in cell division and the cell cycle. The last hub gene cluster contained 11 genes (*CD34*, *CD36*, *CD38*, *CD40*, *CD44*, *CD68*, *CD83*, *CXCL10*, *LOC102185698*, *LOC102186356*, and *PXDN*) from the downregulated DEG sets of F-12 vs. N-12, which are involved in immunity.

### 3.6. WGCNA Analyses

The transcriptome data network at four timepoints for FQs and NBYs, which filtered out the low-abundance (FPKM < 1) and low-variability (FPKM *CV* > 0.5) genes, was constructed based on weighted gene coexpression network analysis (WGCNA). Here, a total of 4038 genes were selected for WGCNA, and 8 modules were identified based on the whole transcriptome profile (Figure 6A), each of which comprised 40–3330 genes (Appendix A). Next, the associations between eight modules and four timepoints for FQs and NBYs were evaluated, and seven modules were identified. More precisely, MEblue (*r* = 0.77, *p* = 0.03) in F-E3, MEblack (*r* = 0.96, *p* = 1 × 10^−4^) in F-0, MEgreenyellow (*r* = 0.98, *p* = 3 × 10^−5^) in F-3, MEbrown (*r* = 0.72, *p* = 0.05) in F-12, MEgreen (*r* = 0.97, *p* = 7 × 10^−5^) and MEmagenta (*r* = 0.85, *p* = 0.008) in N-0, and MEyellow (*r* = 0.99, *p* = 7 × 10^−6^) in N-E3 (Figure 6B).

In addition, to explore the biological functions of the eight modules, GO and KEGG enrichment analyses were carried out. In MEblue, 3330 genes with enriched GO terms (GO:0000786, GO:0008017, GO:0005201, GO:0005518, GO:0034501, GO:0000400, GO:0016477, GO:0003896, and GO:0072137) and KEGG pathways (ko04110 and ko03030) were enriched in the biological process of cell division and proliferation (Figure 7A,B). Moreover, KEGG pathways (ko04151, ko04015, ko04810, ko04392, and ko04371) involved in the regulation of cell proliferation were also enriched (Figure 7B). To further screen the key genes, PPI analysis was performed on the genes involved in the abovementioned GO terms and KEGG pathways in MEblue, and 437 genes were clustered into four categories by k-means clustering (Appendix A). The first cluster (146 nodes and 606 edges) was involved in collagen synthesis in the extracellular matrix of skeletal muscle (Appendix A), and 10 hub genes (*COL6A2*, *COL5A2*, *FBN1*, *COL4A6*, *COL2A1*, *COL11A1*, *COL16A1*, *SERPINH1*, and *COL5A3*) were screened (Figure 7C). The second cluster (117 nodes and 776 edges) included structural constituents of chromatin and nucleosomes (Appendix A), and 10 hub genes (*KIF11*, *AURKB*, *CENPF*, *TTK*, *KIF23*, *NUSAP1*, *PRC1*, *KIF22*, *KIF20A*, and *KIF4A*) were screened (Figure 7D). The third cluster (71 nodes and 107 edges) included genes involved in cell-regulated signaling pathways (Appendix A), and 8 hub genes (*PLCG1*, *PLCB3*, *CALML4*, *ITPR3*, *RYR3*, *RIC8A*, *RGS6*, and *CASQ2*) were screened (Figure 7E). The last cluster (67 nodes and 323 edges) was involved in cilium assembly and transport (Appendix A), and 10 hub genes (*IFT172*, *IFT52*, *IFT140*, *IFT80*, *TFT20*, *KIF3A*, *DYNC2H1*, *BBS1*, *BBS7*, and *BBS2*) were screened (Figure 7F). These results indicated that the *Longissimus dorsi* muscle cells of FQs maintained an active proliferative state at E3.

The MEyellow module, which was significantly positively correlated with N-E3 (*r* = 0.99, *p* = 7 × 10^−6^), contained 128 genes. The GO enrichment analysis revealed a negative regulation of cell population proliferation (GO:008285) and a negative regulation of the BMP signaling pathway (GO:0030514), indicating that the differentiation and proliferation of myoblasts gradually progressed (Figure 8A,B). Moreover, fat cell differentiation was initiated (GO:0050873, GO:0050872), which was verified by the enrichment of a large number of lipid metabolism (GO:0008203, ko04923, ko04975, ko00600, and ko00565) and regulatory signaling pathways (ko03320, ko04152, ko04920, ko04350, and ko04151). The 104 genes in the MEyellow module were used for further PPI analysis (Appendix A), and the 10 hub genes (*COL1A1*, *FMOD*, *CEBPA*, *COL1A2*, *PLIN1*, *PPARG*, *FABP4*, *ADIPOQ*, *LIPE*, and *CIDEC*) are shown in Figure 8C. 

The MEblack module, which was significantly positively correlated with F-0 (*r* = 0.96, *p* = 1 × 10^−4^), clustered 113 genes. The significant enrichment of ribosome-function-related GO terms (GO:0032986, GO:0005840, GO:0003735, and GO:0019843) implied an active protein translation process (ko01230, ko00260, ko00520, and ko00220), which is required for complete myofiber enlargement (GO:0003012, GO:0045662, GO:0046716, and GO:0055001) after differentiation and proliferation (Figure 9A,B). Moreover, the 93 genes in the MEblack module were subjected to further PPI analysis (Appendix A), and 10 hub genes (*RPS2*, *RPSA*, *RPL5*, *RPL23*, *RPL8*, *RPL4*, *RPS15*, *RPS6*, *RPS8*, and *RPS14*) were screened (Figure 9C).

The MEgreen module (*r* = 0.97, *p* = 7 × 10^−5^), which clustered 132 genes, and the MEmagenta module (*r* = 0.85, *p* = 0.008), which clustered 47 genes, were significantly positively correlated with N-0. First, we focused on the GO terms and KEGG pathways that were significantly enriched in the MEgreen module (Figure 10A,B), and insulin-mediated glucose metabolism attracted our attention. The significant enrichment of insulin resistance (ko04931) and glucose homeostasis (GO:0042593) indicated a stable energy source for *Longissimus dorsi* muscle myofibers in the N-0 group. The FoxO signaling pathway (ko04068) and AMPK signaling pathway (ko04152) revealed the metabolic pathway of glucose in myofibers (Figure 10B). On the other hand, a large number of cancer-related pathways (ko05213, ko05216, ko05210, ko05223, and ko05222) were significantly enriched (Figure 10B), and the p53 signaling pathway (ko04115), which restricts abnormal cell proliferation, was also significantly enriched, which may mean that skeletal muscle cell division was further restricted, but the translation of intracellular protein-expressed genes was not as active as that of F-0. Moreover, the 111 genes in the MEblack module were subjected to further PPI analysis (Appendix A), and 8 hub genes (*CDKN1A*, *FOXO1*, *FOXO3*, *RRS1*, *RPP38*, *GADD45A*, *GABARAPL1*, and *ACOX1*) were screened (Figure 10E). Next, we examined the MEmagenta module and found that most of the significantly enriched GO terms (ko05150, ko04610, ko05133, ko05152, and ko05322) and KEGG pathways (GO:0032729, GO:0001772, GO:0030369, and GO:0042608) were related to immunity and diseases (Figure 10C,D). Interestingly, the significant enrichment in the homeostasis of the number of cells within a tissue (GO:0048873) further confirmed the regulatory role of the organism in skeletal muscle cell division in N-0 (Figure 10C). Similarly, the 46 genes in the MEmagenta module were subjected to PPI analysis (Appendix A), and 10 hub genes (*TYROBP*, *ITGB2*, *C1QA*, *LAPTM5*, *C3AR1*, *C1QB*, *CTSS*, *MYO1F*, *CD53*, and *NCKAP1L*) were screened (Figure 10F).

The MEgreenyellow module, which was significantly positively correlated with F-3 (*r* = 0.98, *p* = 3 × 10^−5^), clustered 44 genes, and the gene functional features (Figure 11A,B) appeared to be related to the defense response to viruses (GO:0051607). The PPI network of the MEgreenyellow module was constructed from 37 genes (Figure 11C), and 10 hub genes (Figure 11D) were screened (*ISG15*, *IFIH1*, *RSAD2*, *MX1*, *MX2*, *OAS1*, *DDX58*, *EIF2AK2*, *PARP9*, and *IFIT3*).

The MEbrown module, which was significantly positively correlated with F-12 (*r* = 0.72, *p* = 0.05), contained 204 genes. The synthesis of long-chain fatty acids, which are essential for lipogenesis and deposition in the body through the gluconeogenesis of pyruvate and glycogenic amino acids, is an important pathway of glucose metabolism in ruminants. Interestingly, gluconeogenesis-related metabolic pathways (ko00010, ko00620, ko01200, ko00640, and ko05012) and GO terms (GO:0006094, GO:0004807, and GO:0009931) were strongly enriched in the MEbrown module (Figure 12A,B), and the glucagon signaling pathway (ko04922) may play an important regulatory role. Furthermore, the 163 genes in the MEbrown module were used to construct a PPI network (Appendix A), and 10 hub genes (*ATF3*, *FOSB*, *EGR1*, *EGR3*, *NR4A1*, *DUSP1*, *NR4A2*, *BTG2*, *NR4A3*, and *TRIB1*) were screened (Figure 12C).

### 3.7. Trend Analysis of DEGs at Four Timepoints between FQs and NBYs

A total of 4805 genes were selected and classified into six clusters (Figure 13A). Clusters *K1*, *K2*, and *K5*, which included 3252 genes, were more highly expressed at the E3 timepoint than at 0M, 3M, and 12M in both FQs and NBYs. The expression of 1723 genes in *K1* tended to change more gradually than that of those in *K2* and *K5* during the four growth stages, and genes in *K2* were more highly expressed in FQs than in NBYs after birth. In contrast, genes in *K3* (645 genes) and *K4* (227 genes) were expressed at lower levels at the E3 timepoint than at 0M, 3M, and 12M, and genes in *K3* were more highly expressed in NBYs than in FQs after birth. The expression of genes in *K6* (682 genes) gradually increased with age in FQs, but the expression level did not change significantly over the four developmental stages in NBYs.

KEGG analysis was used to discover the metabolic pathways associated with the differentially expressed genes (Figure 13B). In *K1*, we discovered that many lipid metabolism pathways such as the PPAR signaling pathway (ko03320), cholesterol metabolism (ko04979), fatty acid degradation (ko00071), and AMPK signaling pathway (ko04152) were enriched. *K2* and *K5*, genes highly expressed at the E3 timepoint, were involved in the cell cycle (ko04110). Moreover, many genes in clusters *K3* and *K4* were involved in glycometabolism, such as glycolysis/gluconeogenesis (ko00010), carbon metabolism (ko01200), and fructose and mannose metabolism (ko00051). Genes in *K6* may be associated with abnormal cell proliferation as a large number of cancer-related pathways were enriched. These results indicate that differences in cell proliferation, glycometabolism, and lipid metabolism at the transcriptome level in skeletal muscle are the key factors determining differences in meat production between FQs and NBYs.

To explore the main terms associated with goat production traits, the functions of DEGs in six clusters were classified based on the GO database, and the top five terms of BP (Figure 13C), CC (Figure 13D), and MF (Figure 13E) are listed in Figure 13. We obtained similar results to those of the KEGG enrichment analysis, namely a large number of GO terms related to the cell cycle, glycometabolism, and lipid metabolism. The genes included in the different clusters were mainly involved in the BP skeletal system morphogenesis (GO:0048705), myoblast fusion involved in skeletal muscle regeneration (GO:0014905), mitotic cytokinesis (GO:0000281), the regulation of the glucose metabolic process (GO:0010906), the glycolytic process (GO:0006096), gluconeogenesis (GO:0006094), the very long-chain fatty acid biosynthetic process (GO:0042761), and the negative regulation of the triglyceride catabolic process (GO:0010897). The genes associated with CC included the myosin complex (GO:0016459), collagen trimer (GO:0005581), glycerol-3-phosphate dehydrogenase complex (GO:0009331), and lipid droplet (GO:0005811). The major MF-related genes were DNA binding (GO:0003677), DNA polymerase binding (GO:0070182), and carbohydrate binding (GO:0030246).

### 3.8. qPCR Validation

The results of the qPCR analysis of the expression of 12 genes (*MYH1*, *MYH2*, *MYH4*, *MYH7*, *loc102183280*, *TBC1D*, *CYP1A1*, *FASN*, *IGF2*, *SCPEP1*, *SLN*, and *TPT1*) with different FPKM value ranges involved in muscle development and the RNA-Seq data are shown in Figure 14. The expression levels of genes determined by qPCR and RNA-Seq were highly correlated (Pearson’s correlation coefficient = 0.86), thus validating the RNA-Seq results (Figure 14).

## 4. Discussion

### 4.1. Selection of Endogenous Genes for qPCR Verification 

RNA-Seq analysis is an efficient means of obtaining a genome-wide view of transcript profiles across physiological states. However, qPCR remains the chosen method for high-precision mRNA abundance analysis and is used for the validation of RNA-Seq data. Essential for the reliability of qPCR data is normalization using appropriate internal reference genes, which is now, more than ever before, a fundamental step for accurate gene expression profiling. With the increase in the number of physiological groups in research programs, such as breeds, developmental stages, and organizations, many studies have shown that traditional reference genes, such as *ACTB* and *GAPDH*, are not suitable for target gene normalization [40,41]. In this study, 16 candidate reference genes (Appendix A) were selected from previous reports [36,38,39], 9 of which, including *GAPDH*, were DEGs according to our RNA-Seq data analysis. *API5* showed the least variation and was recommended as the best reference gene, following the results obtained for porcine skeletal muscle at 26 different developmental stages by Niu [40]. 

### 4.2. Myogenesis Progresses More Slowly in FQs Than in NBYs

In mammals, postnatal muscle growth is largely determined by the total number of myofiber, which is determined during prenatal skeletal muscle development [43]. However, myogenesis differs between breeds of the same species at the embryonic stage and largely determines the growth performance and meat quality traits of livestock [44,45,46,47]. Previous studies have shown that myogenesis starts earlier but progresses more slowly in Chinese indigenous pigs (Langtang and Tongcheng) than in exotic pigs (Landrace and Yorkshire), which is considered the main reason that Chinese indigenous pig breeds and exotic species vary greatly in terms of muscle production and performance traits [44,45]. Moreover, the proliferation and differentiation potential of skeletal muscle satellite cells from new myotubes were greater in Nanjiang goats (a Chinese indigenous goat breed) than in Boer goats (a premier meat breed worldwide), possibly contributing to the difference in meat production between the two breeds [46]. On the other hand, primary muscle fibers complete the fusion process and fully form between the 90-day fetal and 120-day fetal stages [6], and some genes associated with skeletal muscle hypertrophy gradually change between the 120-day fetal and 135-day fetal stages in goats [14].

In this study, the expression of genes related to myoblast fusion, skeletal muscle satellite cell proliferation, and growth factors, including *MYOD1* [9], *MYF5* [45], *PAX7* [48], and *MSTN* [49], was greater in FE3 than in NE3, even though the transcript levels of several other myogenesis-related genes (*MEF2A* [48] and *IGF2* [48]) were similar (Figure 15). Interestingly, many downregulated DEGs (*KIF23*, *CCDC69*, *CCNA2*, *MKI67*, *KIF11*, *RACGAP1*, *NUSAP1*, *SKP2*, *ZBTB18*, *NES*, *LOC102180034*, *CAPN6*, *TUBA1A*, *LOC102178700*, and *PEG10*) from the FE3 vs. NE3 set were significantly enriched in the cell cycle (ko04110) and DNA replication (ko03030) terms (Figure 16), which indicated that the differentiation and fusion of myogenesis between FQs and NBYs were not consistent and that the myogenesis of FQs progressed more slowly in FQs than in NBYs. Moreover, the categories “translation, ribosomal structure and biogenesis (7~14.58%)” and “signal transduction mechanisms (6~12.5%)” were highly enriched according to COG classifications, which included the 163 highly expressed DEGs (FPKM > 10) of FE3 vs. NE3 (Figure 17A). The results suggested that differences in key signaling genes determined differences in translation levels in skeletal muscle, which confirmed that the progression of myogenesis was not consistent between FQs and NBYs at E3.

Skeletal muscle requires a great deal of energy and proteins to develop [50]. It has been reported that skeletal muscle hypertrophy requires an increase in the rate of protein synthesis, and one way in which this can be achieved is by increasing the translational capacity of muscle through ribosome biogenesis [51]. In our study, genes encoding ribosomal proteins (*MRPS7*, *RPS8*, *RPL6*, *RPL4*, *RPS11*, *RPS10*, *RPL5*, *RPS6*, *RPL8*, *RPS13*, *RPS24*, *RPS15*, and *RPL23*) were downregulated in the F0 vs. N0 comparison (Figure 15) and enriched in the ribosome (ko03010), which seems to indicate that F0 needs to translate more proteins to support muscle development (Figure 18). Furthermore, the categories “translation, ribosomal structure and biogenesis (13~27.66%)” and “energy production and conversion (12~25.53%)” were highly enriched according to COG classifications, which included the 154 highly expressed DEGs (FPKM > 10) of F0 vs. N0 (Figure 17B).

### 4.3. Insulin Resistance Caused by Low Expression of IRS2 May Determine the Slow Growth Rate of FQs after Weaning

Goats grow quickly and need more energy and many proteins to build muscle fibers at the weaning stage. In the Hainan Black goat, many energy metabolism-related genes and many encoding proteins for muscle contraction were derived from LD muscle at 2 months old [9], and similar results were obtained in the Anhui White goat [14]. Glucose and lipids are the main energy sources of skeletal muscle, so glucose and lipid metabolism play important roles in the growth of skeletal muscle. *IRS2* is a major component of the insulin/insulin-like growth factor pathway [52]. In mice, the deletion of *IRS2* causes peripheral insulin resistance and leads to overt diabetes [53]. Previous research revealed that low-birth-weight goat kids had insulin resistance and decreased hepatic *IRS2* expression; lipid levels were increased in the plasma and skeletal muscle, and lipid accumulation occurred in the skeletal muscle, indicating the impairment of key signaling pathways involved in the regulation of lipid metabolism [54]. The results confirmed the importance of differential glucose and lipid metabolism in goat skeletal muscle for goat growth.

We found that *IRS2* was the unique DEG between FQs and NBYs at four timepoints and was expressed at low levels in FQs (Figure 15). Moreover, insulin resistance (ko04931) was enriched at four timepoints, revealing differences in glucose and lipid metabolism in skeletal muscle between FQs and NBYs (Figure 16, Figure 18, Figure 19 and Figure 20). At 3M, the significantly enriched KEGG pathway, fatty acid metabolism (ko01212) (Figure 19), reflects the efficient utilization of lipids by skeletal muscle in NBYs and the augmentation of fatty acids within skeletal muscle in FQs. Furthermore, 135 highly expressed DEGs (FPKM > 10) were identified between FQs and NBYs at E3, and the categories “lipid transport and metabolism (6~13.95%)” and “amino acid transport and metabolism (6~13.95%)” were ranked high by COG classifications (Figure 17). We speculate that the efficient utilization of lipids in NBYs promotes amino acid metabolism in skeletal muscle to achieve rapid skeletal muscle hypertrophy.

### 4.4. Differences in Glucose and Lipid Metabolism in LD Determine the Deposition of IMF in FQs and NBYs

Fat deposition is an important aspect of meat quality. The greatest impact on fat storage is exerted by energy intake more than maintenance. When the energy requirements for growth are reduced as the animal approaches maturity, the extra energy will be stored as fat both within the muscle and in subcutaneous fat [55]. Many researchers have observed fat deposition differences in breeds that selection can affect, with larger-framed breeds resulting in carcasses with less fat [56]. In recent years, a number of RNA-Seq analyses of skeletal muscle in goats and sheep have confirmed that the differences in energy and lipid metabolism among breeds increase with age. For example, only four genes from the kid (2 months), youth (9 months), and adult (24 months) periods were associated with lipid metabolism in Jianzhou Big-Eared goats [6]. However, at the same age, in Bandur sheep and local sheep (12 to 19 months old) [57], Chaka sheep and Tibetan sheep (12 months old) [58], Liaoning cashmere goats and Ziwuling black goats (9 months old) [18], Duan goats and NBY goats (12 months old), many of the genes identified were related to glucose and lipid metabolism [20].

Herein, the differences in glucose and lipid metabolism began at the E3 timepoint and continued to increase as growth proceeded in FQs and NBYs (Figure 16, Figure 18, Figure 19 and Figure 20). AMPK is the main cellular energy sensor. Activated following a depletion of cellular energy stores, AMPK will restore energy homoeostasis by increasing energy production and limiting energy waste [59]. Compared to FQs, the upregulated genes *CD36* and *ADIROQR2* promoted the absorption of fatty acids and adiponectin by skeletal muscle cells, and the upregulated genes *ACACA*, *ACACB*, and *CPT1A* promoted the β-oxidation of fatty acids in mitochondria to fuel the cells in NBYs at 3M and 12M (Figure 15, Figure 19 and Figure 20). On the other hand, the upregulated genes *IGF1R* and *IRS2* reduced glucose entry into skeletal muscle cells via the insulin signaling pathway (ko04910) in NBYs at the 3M and 12M timepoints (Figure 15, Figure 19 and Figure 20). Furthermore, the downregulated genes *LDH-A*, *PKM*, *HK2*, *PFKP*, *PCK1*, and *GPI* reduced the rate of glycolysis in the skeletal muscle cells of the NBYs at 3M and 12M (Figure 15, Figure 19 and Figure 20). The downregulated genes *ACACA*, *FASN*, *FADS1*, and *ELOVL6* and the upregulated genes *HADHB*, *ACOK1*, *ACAA2*, and *ACSL4* promoted the β-oxidative degradation of fatty acids rather than the synthesis of long-chain fatty acids in the fatty acid metabolic pathway (ko01212) in NBYs at 3M and 12M (Figure 15, Figure 19 and Figure 20). Interestingly, *FASN* was more highly expressed in the fatty acid synthesis pathway (ko00061), and *ACACB* and *ACSL4* were less strongly expressed in FQs than in NBYs at 3M and 12M (Figure 15, Figure 19 and Figure 20). Overall, in skeletal muscle cells after 3M, FQs tended to use glycolysis to supply energy to synthesize more fatty acids for deposition, while NBYs had greater fatty acid utilization activity, which may be the main factor for the significant differences in IMF phenotypes between the two breeds at 12M.

### 4.5. The Unmeasured Differences in Phenotypic Traits of Growth and Meat Quality between FQs and NBYs Limit Our Conclusions

In this study, the transcriptomic sequencing results identified many candidate genes and pathways that play important roles in the growth and IMF deposition between FQs and NBYs. For example, we speculate that myogenesis progresses more slowly in FQs than in NBYs based on many downregulated DEGs from the FE3 vs. NE3 set being significantly enriched in the pathways of “cell cycle” and “DNA replication”. Unfortunately, we still have major weaknesses regarding the relationship between DEG expressions, which were obtained by RNA-Seq, as well as characteristics of skeletal muscle such as the lack of phenotypic measurements of LD muscle properties in FQs and NBYs. Furthermore, insulin resistance is thought to be an important factor in the differences in growth rate and IMF deposition between FQs and NBYs after weaning in our study. In terms of insulin resistance biochemical parameters, the blood glucose level, blood triglyceride level, and blood cholesterol level increase, whereas the blood HDL cholesterol level decreases, thus contributing to glucose and lipid metabolic syndrome [60], but the blood parameters between FQs and NBYs at the four timepoints are still unknown following this study. Finally, we realized that the different feed intakes between NBYs and FQs could cause differences in growth traits and IMF deposition even when feeding on the same diet. Overall, further studies are needed to investigate the phenotypic differences in growth and meat quality between FQs and NBYs, as the few phenotypic traits measured in this study were not sufficient to support the differential expression results obtained by transcriptome analysis.

## 5. Conclusions

The transcriptomic sequencing results from this study identified 48 candidate genes and significant pathways (cell cycle, ribosomes, AMPK signaling pathway, insulin resistance, insulin signaling pathway, fatty acid metabolism, and glycolysis/gluconeogenesis) that play important roles in the growth and IMF deposition of goat LD tissue at different developmental stages and revealed the different molecular mechanisms regulating the muscle development and IMF deposition in FQs and NBYs. Moreover, the results obtained from this study provide a new field of vision for regulating skeletal muscle growth and development in goats, and the identified candidate genes could be used in the selection process. However, the phenotypic variations caused by these DEGs, such as muscle fiber characteristics, serum indexes, and feed intake, still need to be further studied in order to understand the physiological basis of the difference in meat quality and growth between FQs and NBYs.

## Figures and Tables

**Figure 1 animals-14-01770-f001:**
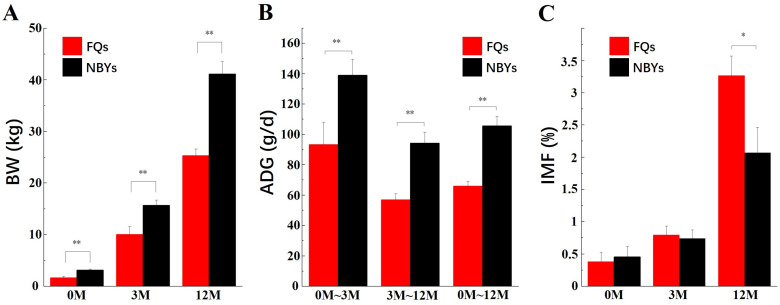
The production traits of FQs and NBYs; ** means *p* < 0.01, * means *p* < 0.05. (**A**) BW. (**B**) ADG. (**C**) IMF.

**Figure 2 animals-14-01770-f002:**
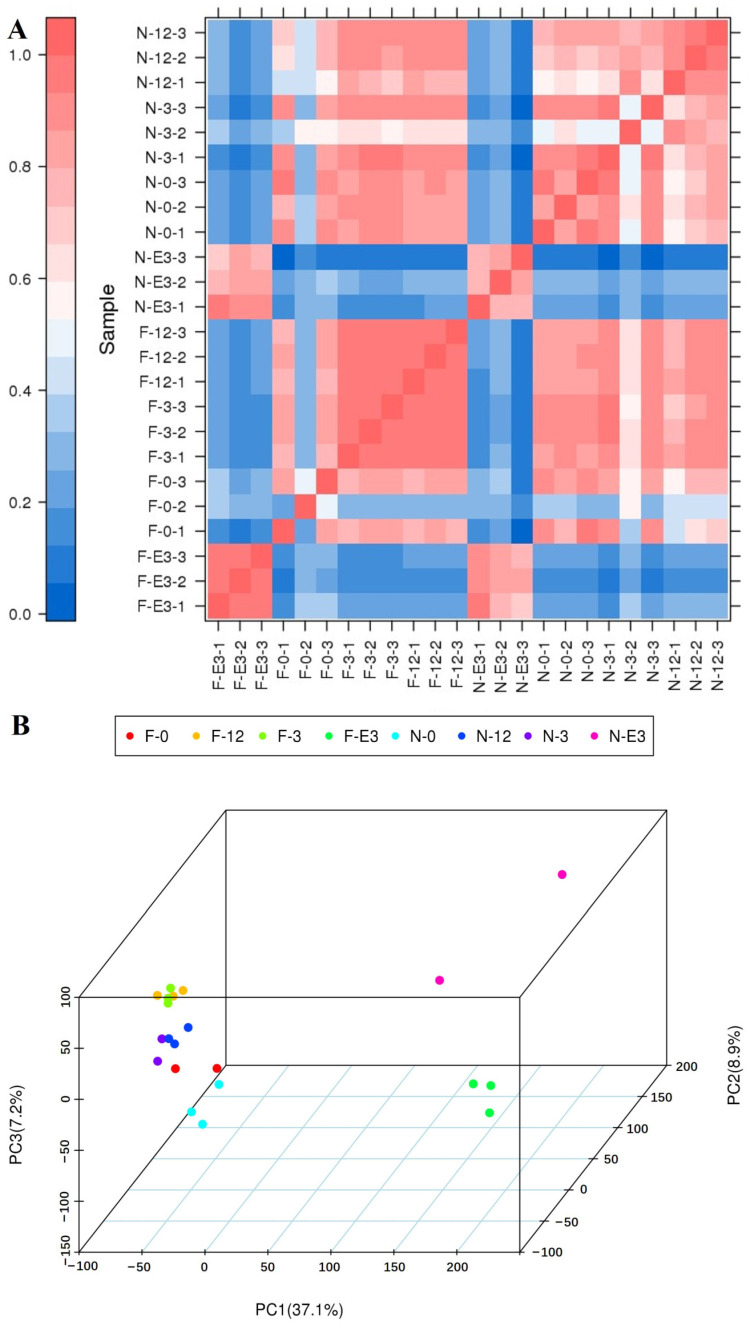
PCC analysis and PCA of FQs and NBYs. (**A**) PCC values for each pair of samples and cluster analysis were performed. (**B**) PCA analysis.

**Figure 3 animals-14-01770-f003:**
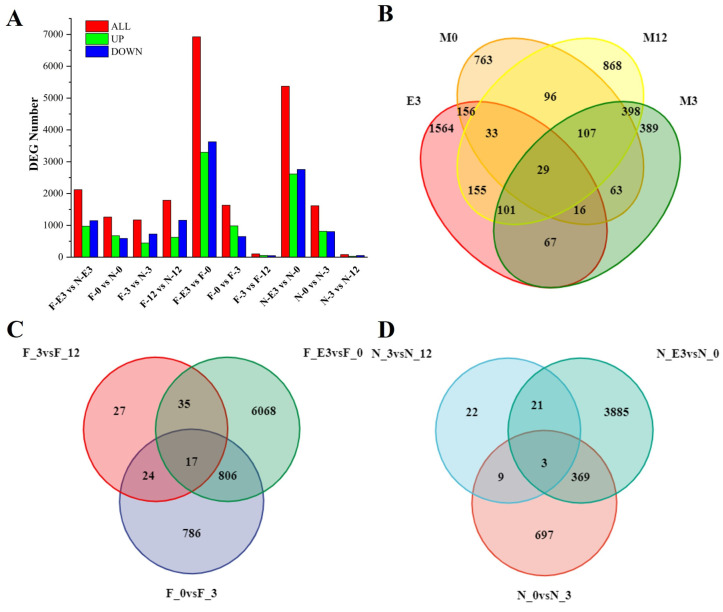
The number of DEGs at four different developmental stages of FQs and NBYs. (**A**) The number of DEGs in different groups. (**B**) Venn plot of DEGs (FQs vs. NBYs) identified at four different developmental stages. (**C**) Venn plot of DEGs at four different developmental stages in FQs. (**D**) Venn plot of DEGs at four different developmental stages in NBYs.

**Figure 4 animals-14-01770-f004:**
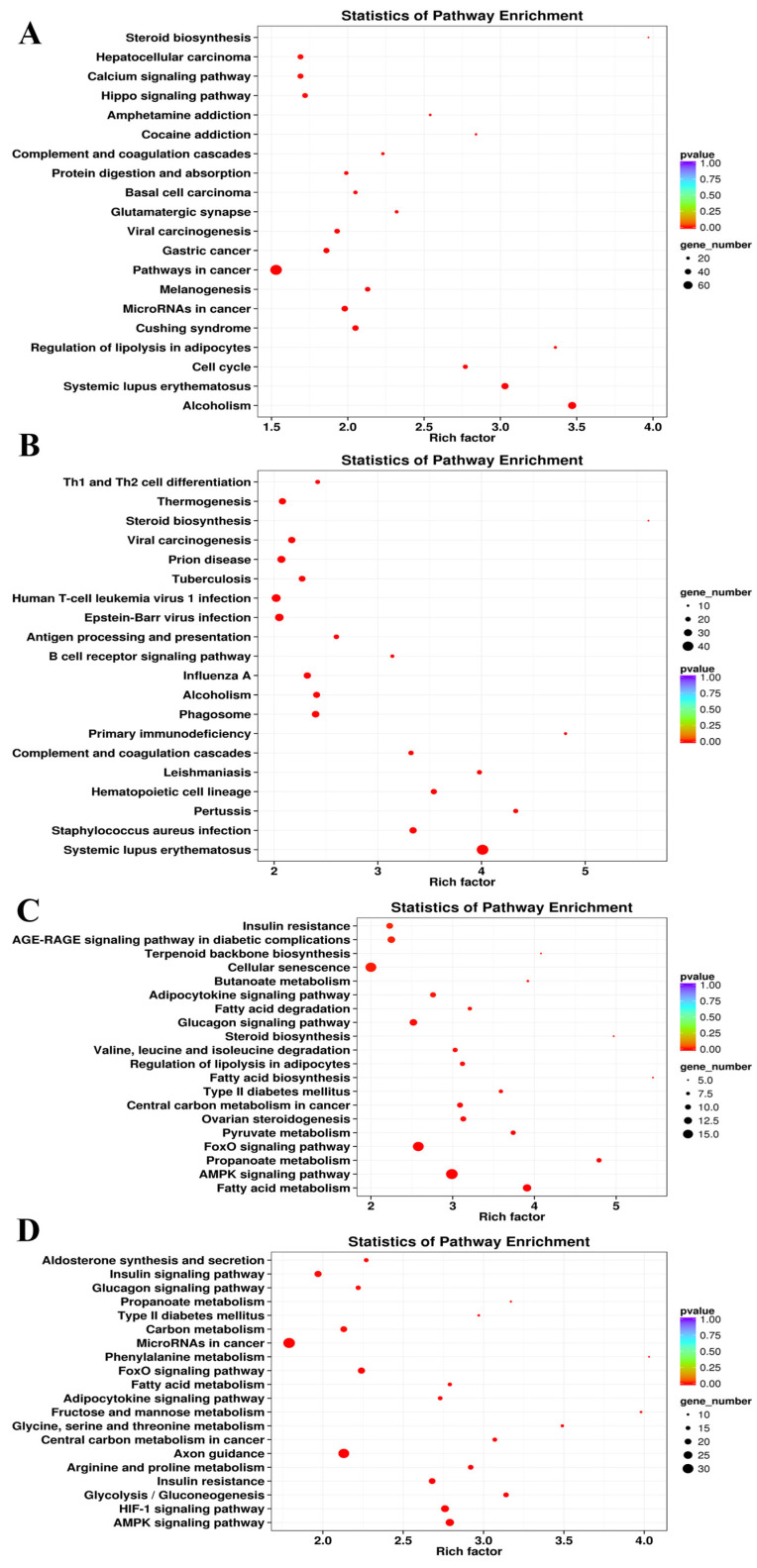
KEGG enrichment analyses at four different developmental stages of FQ and NBY goats. (**A**) F-E3 vs. N-E3. (**B**) F-0 vs. N-0. (**C**) F-3 vs. N-3. (**D**) F-12 vs. N-12.

**Figure 5 animals-14-01770-f005:**
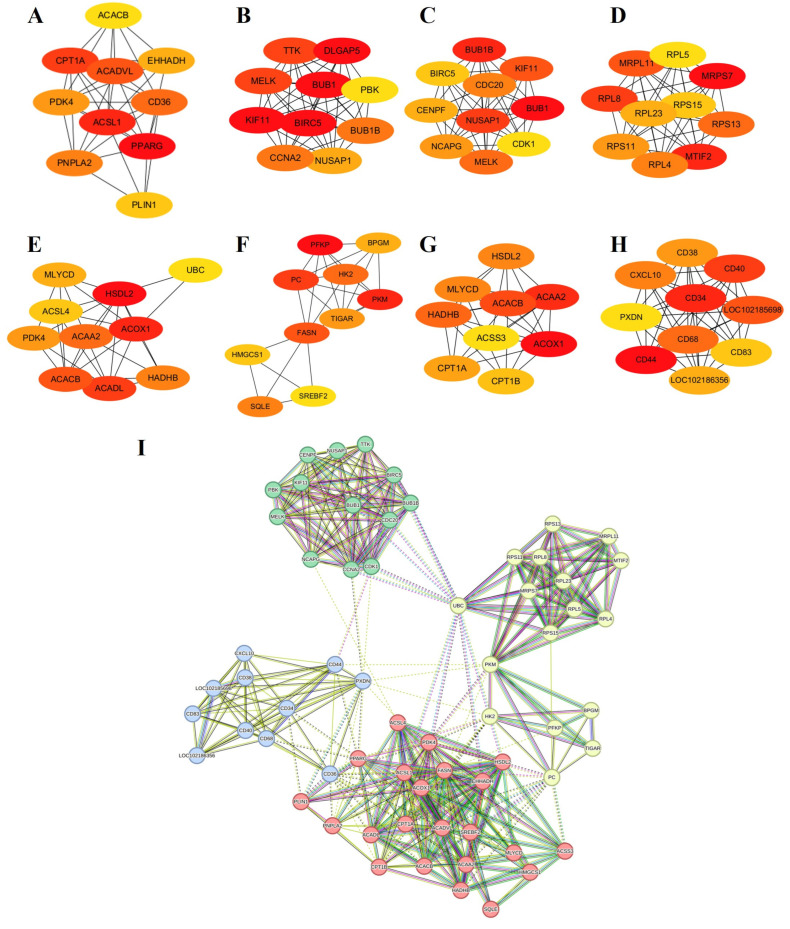
The PPI network of the hub genes at four different developmental stages of FQs and NBYs, and genes with the same color are represented as the same cluster. (**A**) Up DEGs from F-E3 vs. N-E3. (**B**) Down DEGs from F-E3 vs. N-E3. (**C**) Up DEGs from F-0 vs. N-0. (**D**) Down DEGs from F-0 vs. N-0. (**E**) Up DEGs from F-3 vs. N-3. (**F**) Down DEGs from F-3 vs. N-3. (**G**) Up DEGs from F-12 vs. N-12. (**H**) Down DEGs from F-12 vs. N-12. (**I**) PPI network of the hub genes.

**Figure 6 animals-14-01770-f006:**
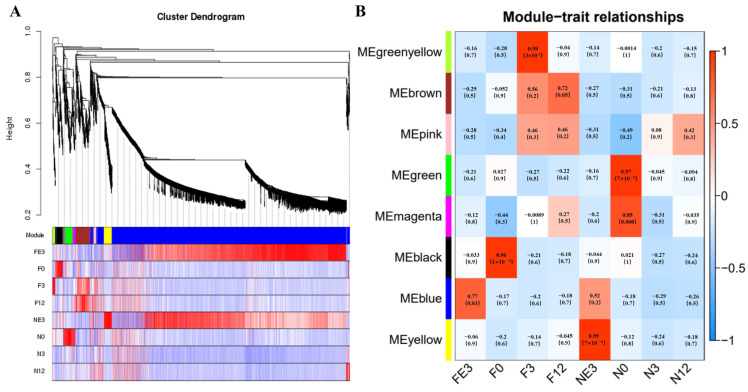
WGCNA of FQs and NBYs at four timepoints. (**A**) Hierarchical cluster trees showing the coexpression modules identified by WGCNA. (**B**) Coexpression modules identified by WGCNA.

**Figure 7 animals-14-01770-f007:**
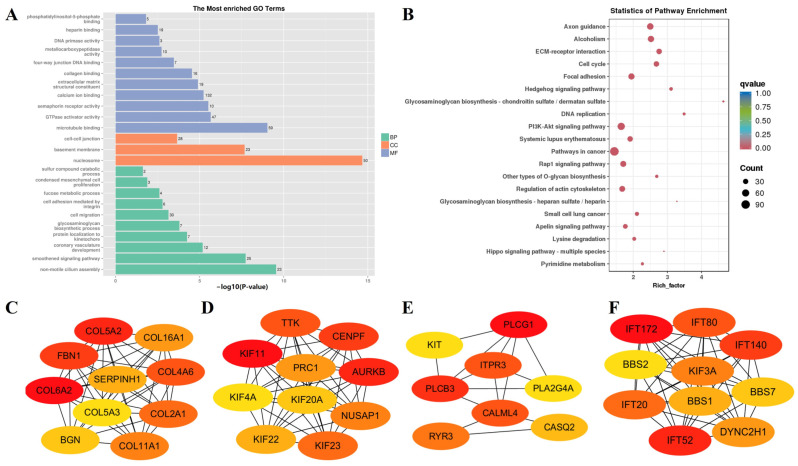
Gene function enrichment analysis of MEblue. (**A**) GO enrichment analysis. (**B**) KEGG enrichment analysis. (**C**) Hub gene network of cluster 1 in MEblue. (**D**) Hub gene network of cluster 2 in MEblue. (**E**) Hub gene network of cluster 3 in MEblue. (**F**) Hub gene network of cluster 4 in MEblue.

**Figure 8 animals-14-01770-f008:**
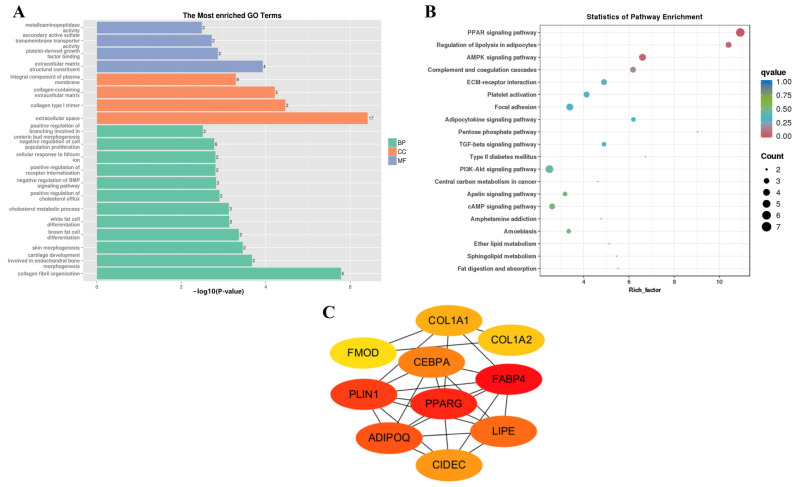
Gene function enrichment analysis of MEyellow. (**A**) GO enrichment analysis. (**B**) KEGG enrichment analysis. (**C**) Hub gene network in MEyellow.

**Figure 9 animals-14-01770-f009:**
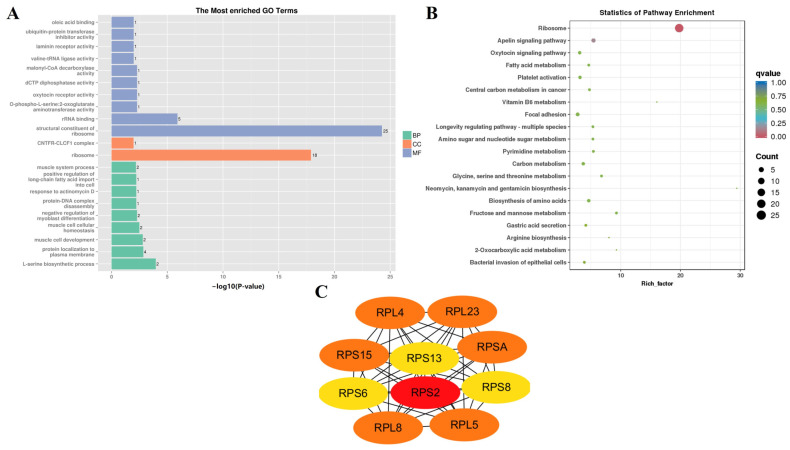
Gene function enrichment analysis of MEblack. (**A**) GO enrichment analysis. (**B**) KEGG enrichment analysis. (**C**) Hub gene network in MEblack.

**Figure 10 animals-14-01770-f010:**
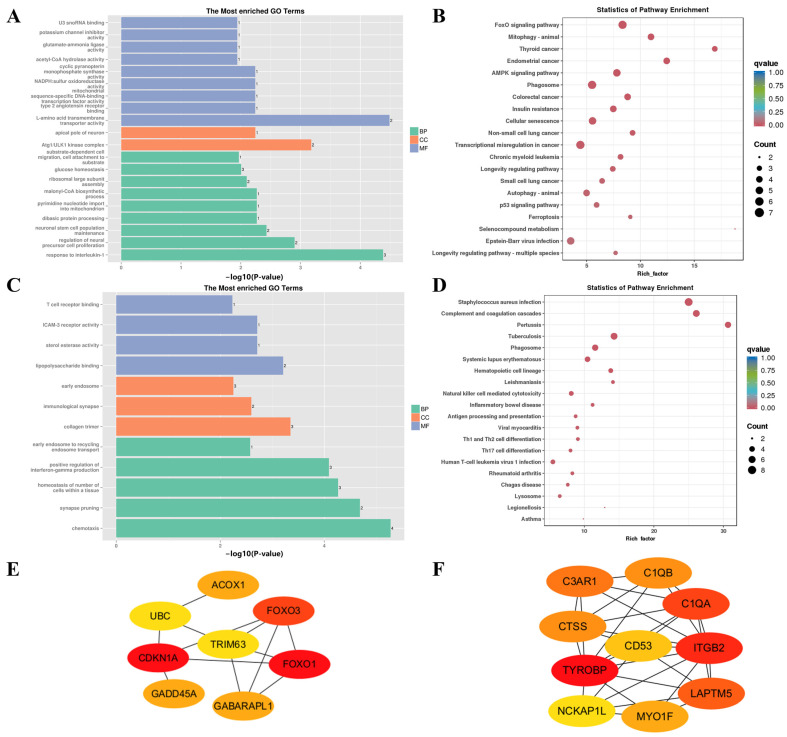
Gene function enrichment analysis of MEgreen and MEmagenta. (**A**) GO enrichment analysis of MEgreen. (**B**) KEGG enrichment analysis of MEgreen. (**C**) GO enrichment analysis of MEmagenta. (**D**) KEGG enrichment analysis of MEmagenta. (**E**) Hub genes network in MEgreen. (**F**) Hub gene network in MEmagenta.

**Figure 11 animals-14-01770-f011:**
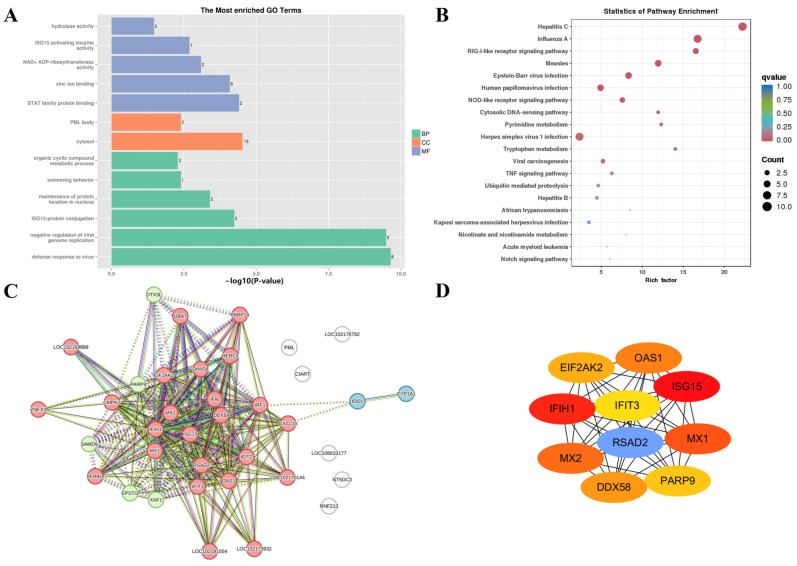
Gene function enrichment analysis of MEyellowgreen. (**A**) GO enrichment analysis of MEyellogreen. (**B**) KEGG enrichment analysis of MEyellowgreen. (**C**) PPI network from MEyellowgreen, and genes with the same color are represented as the same cluster. (**D**) Hub gene network in MEyellowgreen.

**Figure 12 animals-14-01770-f012:**
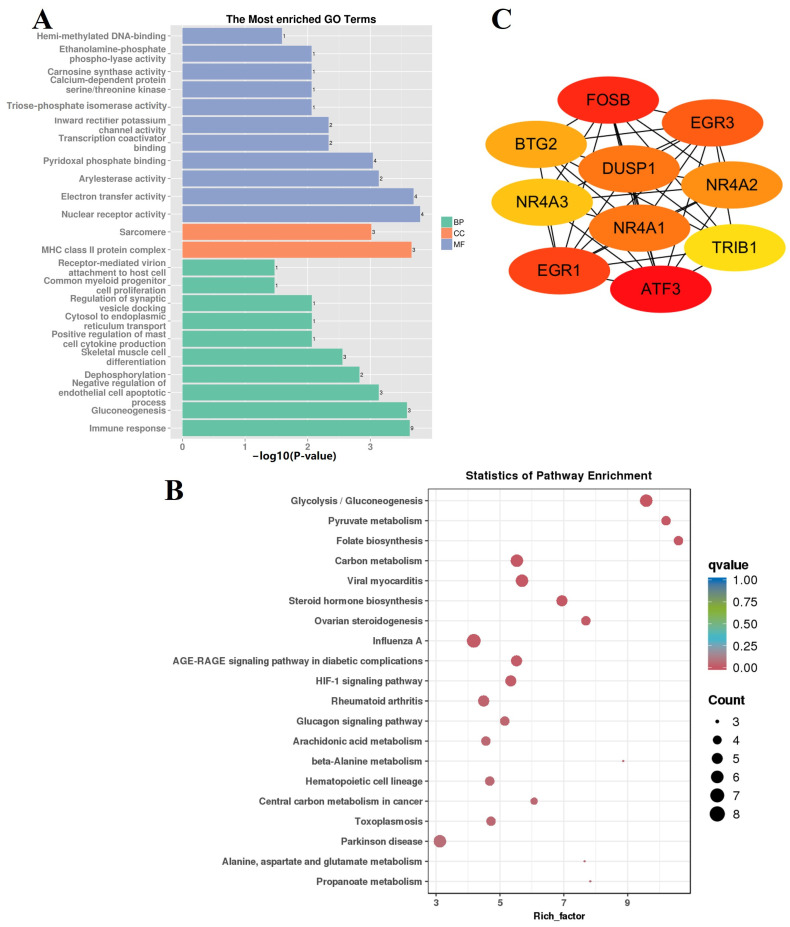
Gene function enrichment analysis of MEbrown. (**A**) Results from GO enrichment analysis of MEbrown. (**B**) KEGG enrichment analysis of MEbrown. (**C**) Hub gene network in MEbrown.

**Figure 13 animals-14-01770-f013:**
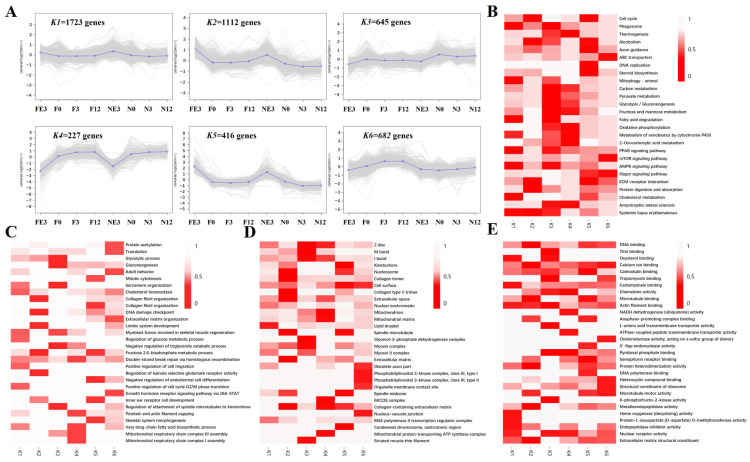
Overview of serial analysis of DEGs identified by pairwise comparisons between FQs and NBYs at E3, 0M, 3M, and 12M. (**A**) The six clusters of DEGs. (**B**) Top enrichment pathways of the DEGs in different clusters according to the KEGG database. (**C**) Top enriched BP terms of the DEGs in different clusters according to the GO database. (**D**) Top enriched CC terms of the DEGs in different clusters according to the GO database. (**E**) Top enriched MF terms of the DEGs in different clusters according to the GO database. The significance of the most represented KEGG pathways and GO terms in each main cluster are indicated using the Q-value (red), and white areas represent missing values.

**Figure 14 animals-14-01770-f014:**
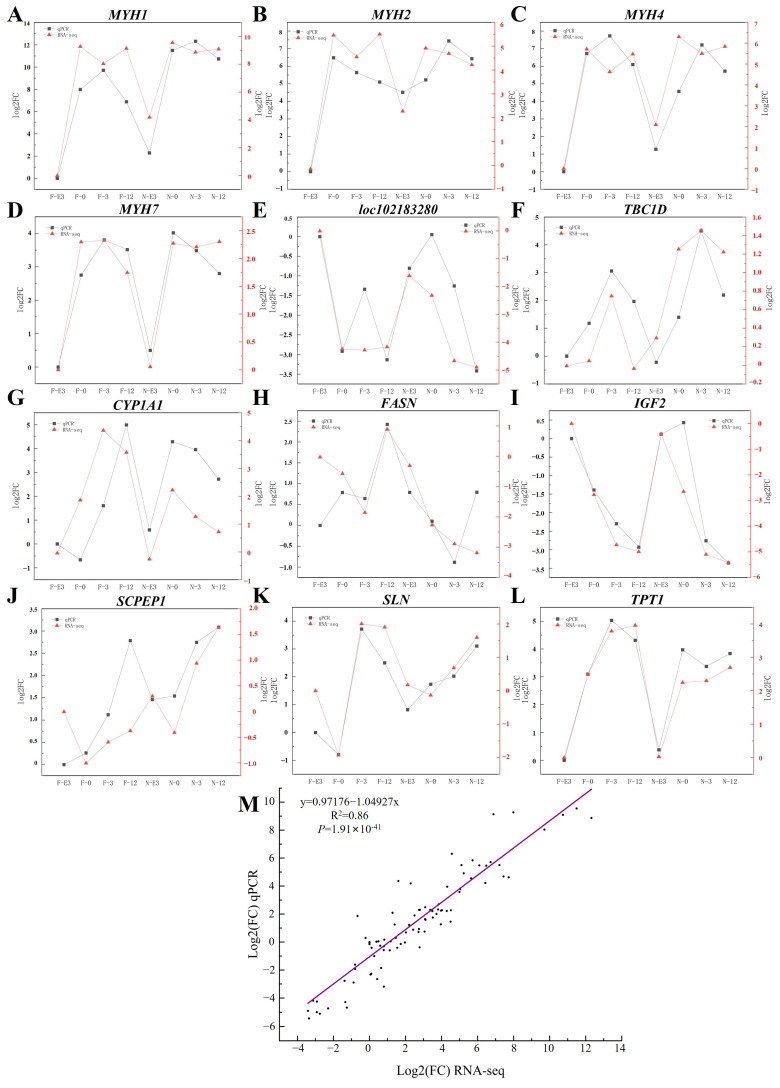
Comparison of the expression levels of 12 genes obtained by RNA-Seq and qPCR detection. (**A**) *MYH1*. (**B**) *MYH2*. (**C**) *MYH4*. (**D**) *MYH7*. (**E**) *loc102183280*. (**F**) *TBC1D*. (**G**) *CYP1A1*. (**H**) *FASN*. (**I**) *IGF2*. (**J**) *SCPEP1*. (**K**) *SLN*. (**L**) *TPT1*. (**M**) Correlation analysis between the RNA-Seq and qPCR data of the expression levels of 12 genes.

**Figure 15 animals-14-01770-f015:**
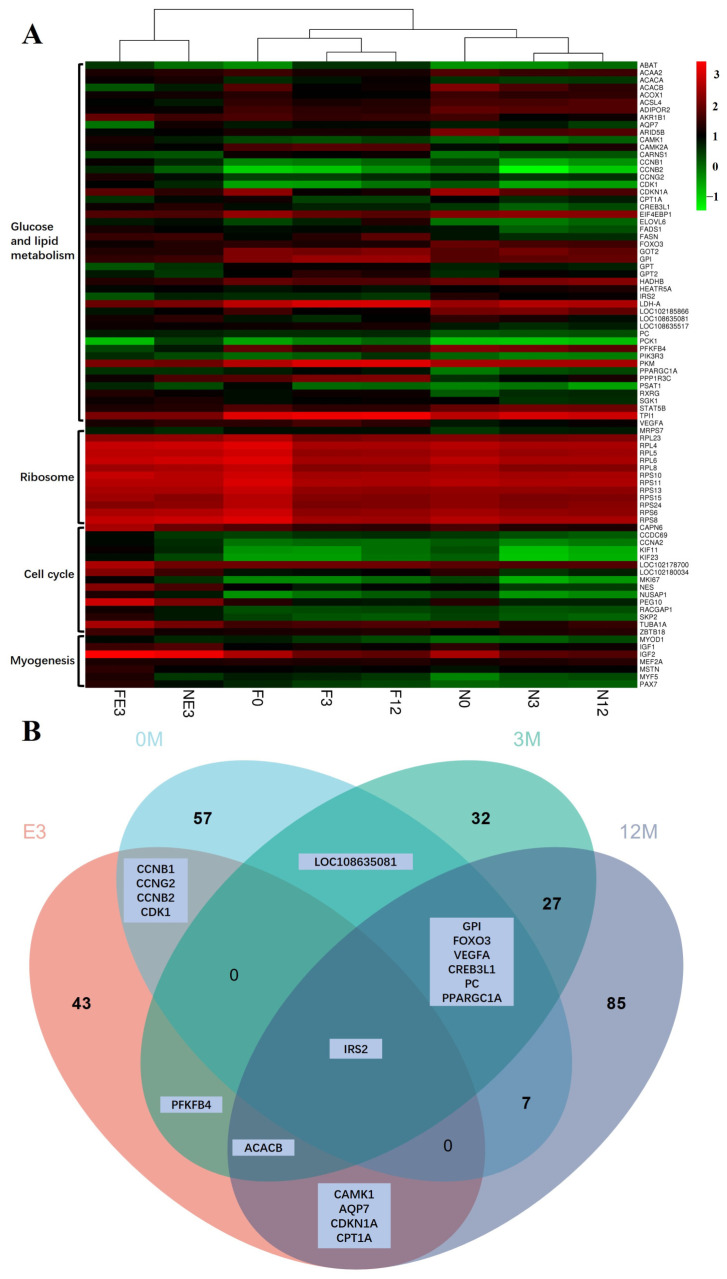
Hot map and Venn diagram of key DEGs at four timepoints. (**A**) Hot map. (**B**) Venn diagram of DEGs related to glucose and lipid metabolism.

**Figure 16 animals-14-01770-f016:**
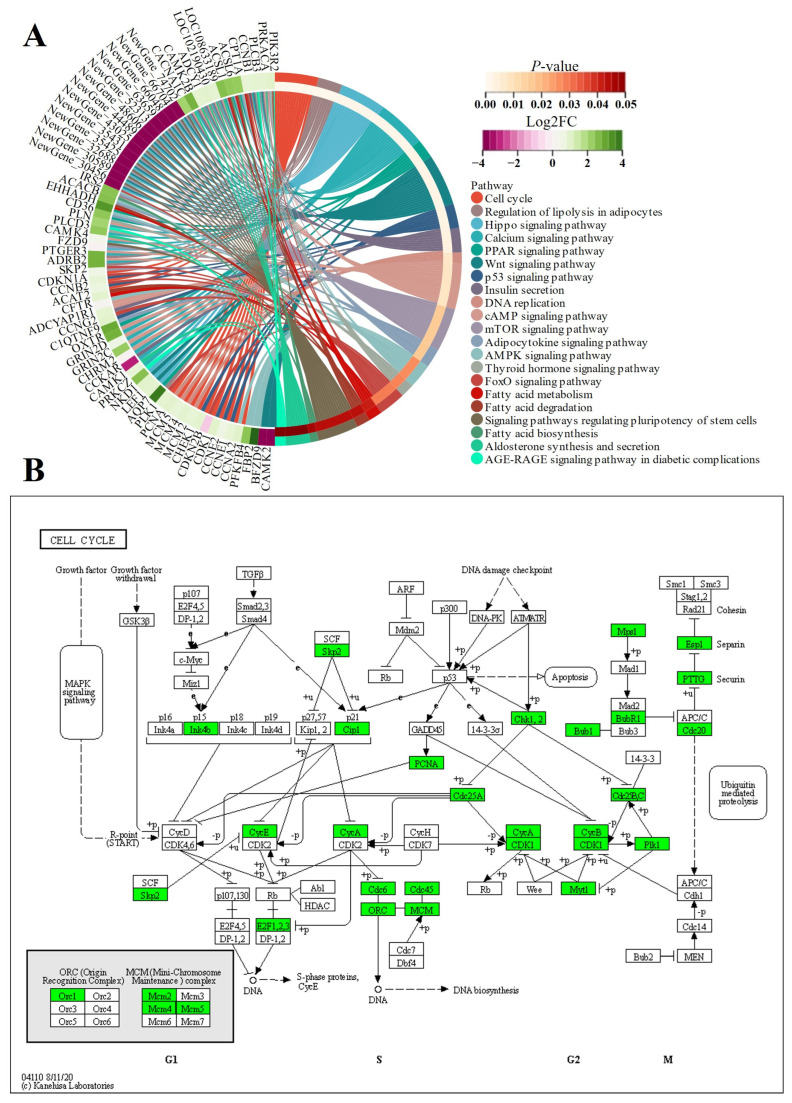
Chord diagram and KEGG pathways from FE3 vs. NE3. (**A**) FE3 vs. NE3 comparison. (**B**) ko04110, which is a subfigure label form http://www.genome.jp/kegg1b.html (accessed on 21 January 2024), in the FE3 vs. NE3 comparison, and downregulated genes were in green.

**Figure 17 animals-14-01770-f017:**
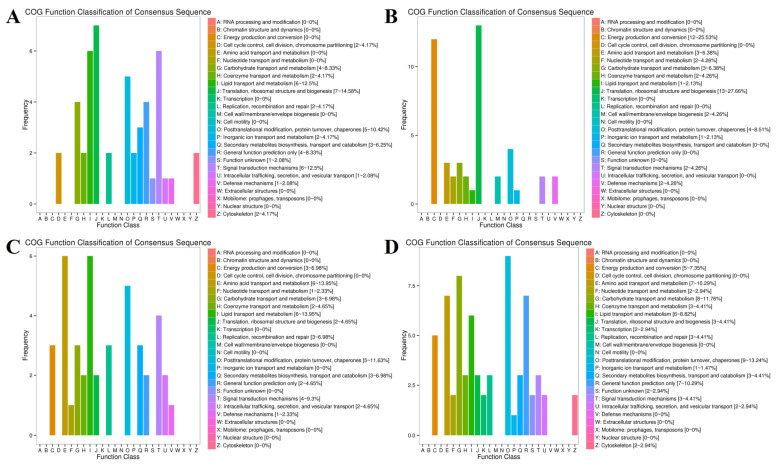
Histogram of COG classification of DEGs with FPKM > 10. (**A**) FE3 vs. NE3. (**B**) F0 vs. N0. (**C**) F3 vs. N3. (**D**) F12 vs. N12.

**Figure 18 animals-14-01770-f018:**
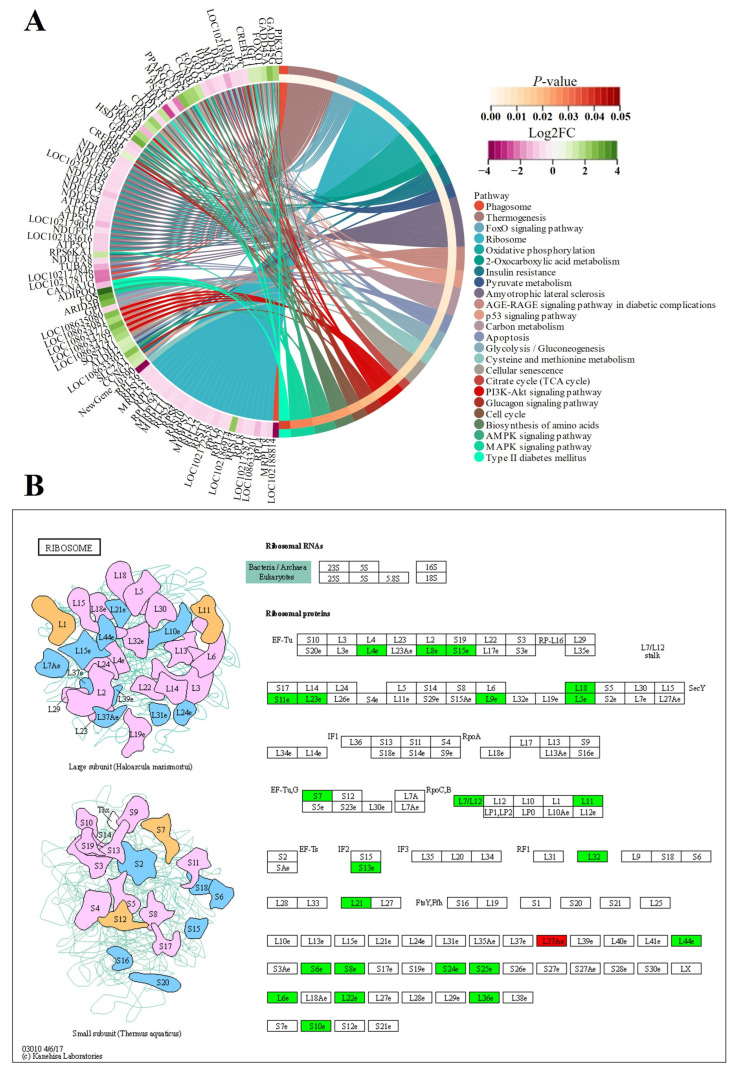
Chord diagram and KEGG pathways from F0 vs. N0. (**A**) F0 vs. N0 set. (**B**) ko03010, which is a subfigure label form http://www.genome.jp/kegg1b.html (accessed on 21 January 2024), in the F0 vs. N0 set, downregulated genes were in green and upregulated gene was in red.

**Figure 19 animals-14-01770-f019:**
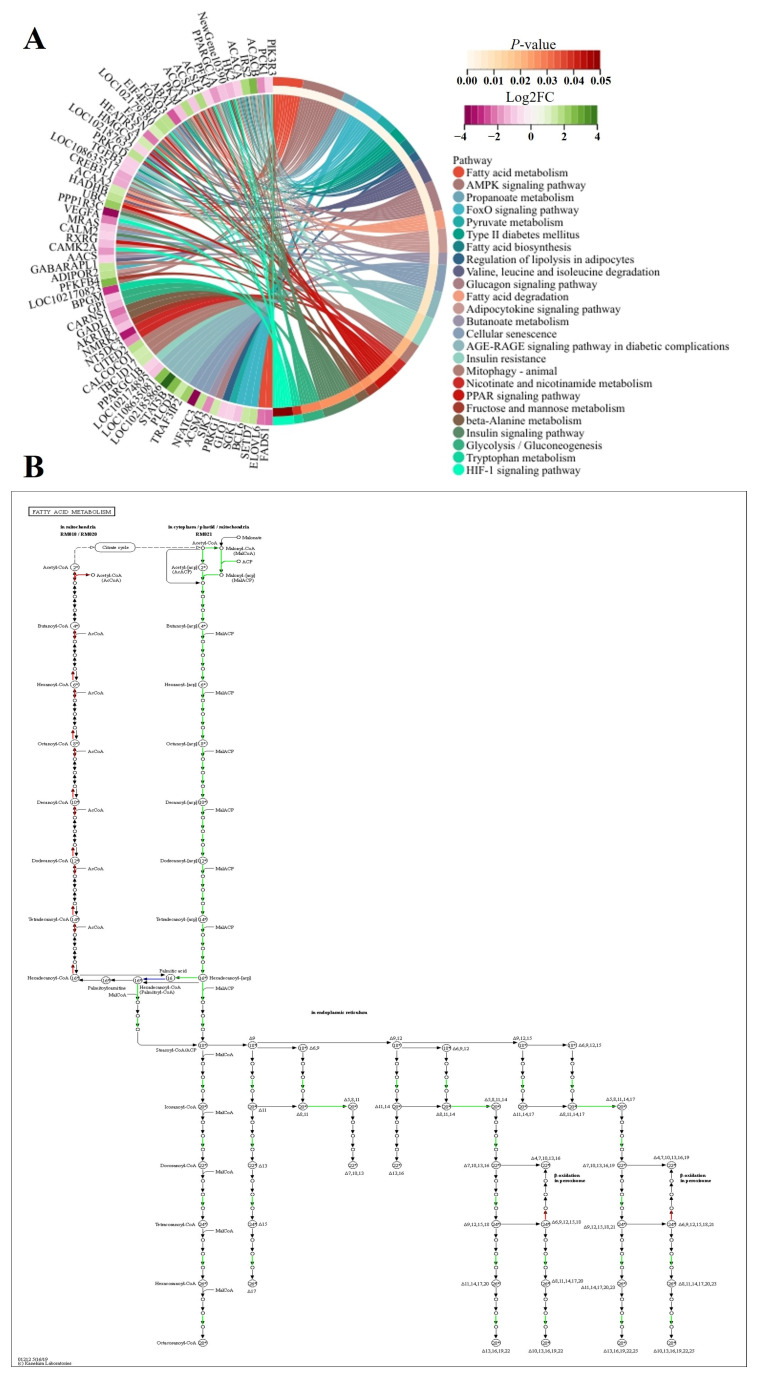
Chord diagram and KEGG pathways from F3 vs. N3. (**A**) F3 vs. N3 set. (**B**) ko01212, which is a subfigure label form http://www.genome.jp/kegg1b.html (accessed on 21 January 2024), in the F3 vs. N3 set, downregulated genes were in green and upregulated genes were in red.

**Figure 20 animals-14-01770-f020:**
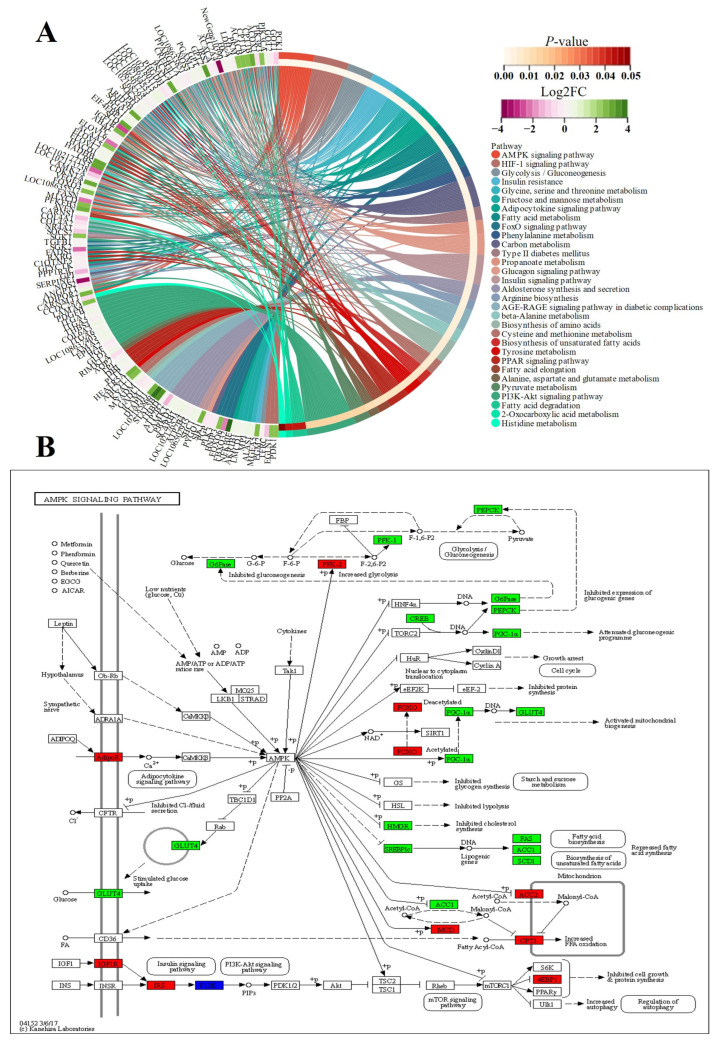
Chord diagram and KEGG pathways from F12 vs. N12. (**A**) F12 vs. N12 set. (**B**) ko04110, which is a subfigure label form http://www.genome.jp/kegg1b.html (accessed on 21 January 2024), in the FE3 vs. NE3 set, downregulated genes were in green and upregulated genes were in red.

**Table 1 animals-14-01770-t001:** GO enrichment of the DEGs from the four timepoints.

Sample Pair	DEGs Number	Number of Significantly Enriched GO			Number of GO Using REViGO		
		BP	CC	MF	BP	CC	MF
F-E3 vs. N-E3	1421	244	92	93	145	56	83
F-0 vs. N-0	1156	284	63	176	158	55	152
F-3 vs. N-3	1121	358	45	113	265	38	99
F-12 vs. N-12	1365	314	41	88	194	31	76

**Table 2 animals-14-01770-t002:** Major GO enrichment of the DEGs from the four timepoints according to the removal of redundant GO terms.

Sample Pair	Class	Annotation	GO ID	*p*-Values
F-E3 vs. N-E3	BP	regulation of attachment of spindle microtubules to kinetochore	GO:0051988	1.55 × 10^−5^
		antibacterial humoral response	GO:0019731	1.63 × 10^−5^
		mitotic cytokinesis	GO:0000281	2.25 × 10^−5^
		CENP-A containing nucleosome assembly	GO:0034080	5.08 × 10^−5^
		myoblast fusion involved in skeletal muscle regeneration	GO:0014905	0.00017
	CC	nucleosome	GO:0000786	1.76 × 10^−19^
		kinetochore	GO:0000776	3.96 × 10^−5^
		I band	GO:0031674	6.42 × 10^−5^
		collagen-containing extracellular matrix	GO:0062023	0.00014
		insulin-like growth factor ternary complex	GO:0042567	0.00038
	MF	protein heterodimerization activity	GO:0046982	1.38 × 10^−16^
		anaphase-promoting complex binding	GO:0010997	0.00030
		glutamate-gated calcium ion channel activity	GO:0022849	0.00296
		fructose-2,6-bisphosphate 2-phosphatase activity	GO:0004331	0.00477
		creatine kinase activity	GO:0004111	0.02604
F-0 vs. N-0	BP	mitochondrial respiratory chain complex III assembly	GO:0034551	8.12 × 10^−6^
		complement activation	GO:0006956	4.97 × 10^−5^
		mitochondrial transcription	GO:0006390	0.00011
		osteoclast differentiation	GO:0030316	0.00024
		apoptotic cell clearance	GO:0043277	0.00432
	CC	nucleosome	GO:0000786	7.46 × 10^−9^
		external side of plasma membrane	GO:0009897	0.00042
		cell surface	GO:0009986	0.00301
		host cell nucleus	GO:0042025	0.00556
		cell–cell junction	GO:0005911	0.00544
	MF	protein heterodimerization activity	GO:0046982	1.76 × 10^−6^
		DNA-binding transcription activator activity, RNA polymerase II-specific	GO:0001228	0.00054
		oxidoreductase activity, acting on a sulfur group of donors	GO:0016667	0.00101
		L-amino acid transmembrane transporter activity	GO:0015179	0.00228
		NAD(P)+-protein-arginine ADP-ribosyltransferase activity	GO:0003956	0.00329
F-3 vs. N-3	BP	negative regulation of chemokine production	GO:0032682	8.31 × 10^−5^
		sterol biosynthetic process	GO:0016126	3.22 × 10^−4^
		B cell apoptotic process	GO:0001783	7.78 × 10^−4^
		outflow tract morphogenesis	GO:0003151	0.00165
		integrin biosynthetic process	GO:0045112	0.00191
	CC	basement membrane	GO:0005604	0.00014
		host cell nucleus	GO:0042025	0.00295
		sarcomere	GO:0030017	0.00329
		membrane raft	GO:0045121	0.03073
		PCSK9-LDLR complex	GO:1990666	0.03997
	MF	sulfonylurea receptor activity	GO:0008281	0.00163
		L-amino acid transmembrane transporter activity	GO:0015179	0.00163
		fibronectin binding	GO:0001968	0.00328
		beta-galactoside (CMP) alpha-2,3-sialyltransferase activity	GO:0003836	0.00478
		medium-chain-acyl-CoA dehydrogenase activity	GO:0070991	0.00478
F-12 vs. N-12	BP	angiogenesis	GO:0001525	2.78 × 10^−7^
		gluconeogenesis	GO:0006094	3.53 × 10^−6^
		negative regulation of myoblast differentiation	GO:0045662	0.00018
		cellular response to leukemia inhibitory factor	GO:1990830	0.00045
		protein localization to plasma membrane	GO:0072659	0.00205
	CC	basement membrane	GO:0005604	5.11 × 10^−8^
		extracellular space	GO:0005615	5.88 × 10^−6^
		M band	GO:0031430	0.00137
		adherens junction	GO:0005912	0.00855
		collagen trimer	GO:0005581	0.01164
	MF	actin monomer binding	GO:0003785	8.29 × 10^−5^
		DNA-binding transcription repressor activity, RNA polymerase II-specific	GO:0001227	0.00012
		L-ascorbic acid binding	GO:0031418	0.00034
		metalloendopeptidase activity	GO:0004222	0.00076
		purine-nucleoside phosphorylase activity	GO:0004731	0.00425

## Data Availability

The data presented in this study are available on request from the corresponding author.

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
