# Peer review of "Temporal Transcriptome Dynamics of *Longissimus dorsi* Reveals the Mechanism of the Differences in Muscle Development and IMF Deposition between Fuqing Goats and Nubian Goats"

_animals, 2024, doi:10.3390/ani14121770_

Round 1

Reviewer 1 Report

Comments and Suggestions for Authors

Review of the article (Temporal Transcriptome Dynamics of Longissimus Dorsi Reveals the Mechanism of the differences in muscle development and IMF deposition between Fuqing Goats and Nubian Goats). We congratulate the authors for their work, but would like to suggest important modifications. We suggest inserting more performance and housing data. This is important for discussing the information found through gene expression. Important changes must be made for the article to be considered for publication.

Author Response

Dear editor and the anonymous reviewers,

We have carefully read the comments and made some modifications to our manuscript submitted to Animals (manuscript ID: 3005337). Below is the response we made one by one to the reviewer’s comments:

[The response to reviewer 1 comments]

[Point 1]

Genetic resources (breeds) should be addressed at the beginning of the introduction. There, differences must be shown through bibliography. Start by demonstrating the differences in performance, meat quality and then differences in gene expression.

[Response 1]

We appreciate your insightful and constructive comments, and we have carefully addressed these concerns and made proper revisions to the manuscript (see the revised section, which is marked with red font on pages 1-2).

[Point 2]

If there are more variables such as dry matter intake, protein intake, serum insulin levels and other variables, we would like to see them added to the study.

[Response 2]

Thank you for your advice, and we agree with your suggestion. Unfortunately, we are unable to provide additional data on these variables, as these were not accounted for in the design of the trial, and we are unable to provide additional measurements of these data as the trial goat were slaughtered.

[Point 3]

Please completely revise your discussion. I found no relationship between the growth characteristics of the kids and the expression of the genes evaluated. Please discuss the differences in growth and development of kids based on possible mechanisms supported by gene expression. Without an important improvement in the discussion, the work will not be considered an important contribution to the journal.

[Response 3]

We appreciate your profound insights and constructive comments. To revise our discussion, we further carried out an analysis of DEGs with high expression levels (Figure 15, page 29) and added references to discuss the differences in the growth and development of kids (3M) based on our results (pages 29-31, red font).

We sincerely hope that the current manuscript upon modification (manuscript ID: 3005337) can meet the requirements of Animals. Don’t hesitate to contact us if there is any question.

Best regards!

Reviewer 2 Report

Comments and Suggestions for Authors

This study constructed the mRNA expression profilings of skeletal muscles fom Fuqing goat breed and Nubia goat breed, which identify some petential genes and pathways related to muscle development and IMF deposition.  It was well disigned. This study will throw light into the mechanisms underlying animal skeleatal muscle development and IMF deposition.

As RNA-Seq was perfomed for the muscle samples from two goat breeds at four developmental stages , WGCNA analysis and Short time-series expression miner (STEM) clustering analysis need to be added, which  can  provide more useful information and identify key genes or pathways.

Comments on the Quality of English Language

To improve the manuscirpt quanlity, Minor rivisons are needed. There are few mistakes in the manuscirpt, such as grammar and tense. 

Author Response

Dear editor and the anonymous reviewers,

We have carefully read the comments and made some modifications to our manuscript submitted to Animals (manuscript ID: 3005337). Below is the response we made one by one to the reviewer’s comments:

[The response to reviewer 2 comments]

[Point 1]

As RNA-Seq was performed for the muscle samples from two goat breeds at four developmental stages, WGCNA analysis and Short time-series expression miner (STEM) clustering analysis need to be added, which  can  provide more useful information and identify key genes or pathways.

[Response 1]

We appreciate the thoughtful review and constructive feedback provided. To improve our manuscript according to your suggestion, we further carried out a trend analysis of DEGs at four timepoints (pages 24-26 with red font) and added a discussion of the WGCNA analysis section in the manuscript (pages 14-24).

[Point 2]

To improve the manuscirpt quanlity, Minor rivisons are needed. There are few mistakes in the manuscirpt, such as grammar and tense. 

[Response 2]

Thank you for your advice. We have made the necessary modifications per your request.

We sincerely hope that the current manuscript upon modification (manuscript ID: 3005337) can meet the requirements of Animals. Don’t hesitate to contact us if there is any question.

Best regards!

Round 2

Reviewer 1 Report

Comments and Suggestions for Authors

The authors made some important adjustments to the introduction and results, but unfortunately we still have major weaknesses in the relationship between gene expression and other variables. I suggest that at the end of the discussion these weaknesses (lack of data on blood parameters, intake, fat deposition...) be reported. The conclusion must make this weakness clear.

Author Response

Dear editor and the anonymous reviewers,

We have carefully read the comments and made some modifications to our manuscript submitted to Animals (manuscript ID: 3005337). Below is the response we made one by one to the reviewer’s comments:

[The response to reviewer 1 comments]

[Point 1] The authors made some important adjustments to the introduction and results, but unfortunately we still have major weaknesses in the relationship between gene expression and other variables. I suggest that at the end of the discussion these weaknesses (lack of data on blood parameters, intake, fat deposition...) be reported. The conclusion must make this weakness clear.

[Response 1]

We are very sorry that we did not make further modification on this comment in the first round of modification, and appreciate your insightful and constructive comments again. This time, we have carefully addressed these concerns and made proper revisions to the manuscript (see the revised section, which is marked with red font on pages 37).

We sincerely hope that the current manuscript upon modification (manuscript ID: 3005337) can meet the requirements of Animals. Don’t hesitate to contact us if there is any question.

Best regards!

Reviewer 2 Report

Comments and Suggestions for Authors

This manuscript has been improved and  meet for publication after revision.

Author Response

Dear editor and the anonymous reviewers,

We have carefully read the comments and made some modifications to our manuscript submitted to Animals (manuscript ID: 3005337). Below is the response we made one by one to the reviewer’s comments:

[The response to reviewer 2 comments]

[Point 1] This manuscript has been improved and meet for publication after revision.

[Response 1]

We sincerely appreciate your recognition of our work !

We sincerely hope that the current manuscript upon modification (manuscript ID: 3005337) can meet the requirements of Animals. Don’t hesitate to contact us if there is any question.

Best regards!